# QUT-DV25: A Dataset for Dynamic Analysis of Next-Gen Software Supply Chain Attacks

**Sk Tanzir Mehedi**[1,*]**, Raja Jurdak**[1]**, Chadni Islam**[2]**, and Gowri Ramachandran**[1]

[1]Queensland University of Technology, Brisbane, Australia
[2]Edith Cowan University, Joondalup, Australia

## Abstract

Securing software supply chains is a growing challenge due to the inadequacy of existing datasets in capturing the complexity of next-gen attacks, such as multiphase malware execution, remote access activation, and dynamic payload generation. Existing datasets, which rely on metadata inspection and static code analysis, are inadequate for detecting such attacks. This creates a critical gap because these datasets do not capture what happens during and after a package is installed. To address this gap, we present QUT-DV25, a dynamic analysis dataset specifically designed to support and advance research on detecting and mitigating supply chain attacks within the Python Package Index (PyPI) ecosystem. This dataset captures install and post-install-time traces from 14,271 Python packages, of which 7,127 are malicious. The packages are executed in an isolated sandbox environment using an extended Berkeley Packet Filter (eBPF) kernel and user-level probes. It captures 36 real-time features, that includes system calls, network traffic, resource usages, directory access patterns, dependency logs, and installation behaviors, enabling the study of next-gen attack vectors. ML analysis using the `QUT-DV25` dataset identified four malicious PyPI packages previously labeled as benign, each with thousands of downloads. These packages deployed covert remote access and multi-phase payloads, were reported to PyPI maintainers, and subsequently removed. This highlights the practical value of `QUT-DV25`, as it outperforms reactive, metadata, and static datasets, offering a robust foundation for developing and benchmarking advanced threat detection within the evolving software supply chain ecosystem.

## 1 Introduction

The exponential growth of Open-Source Software (OSS) has introduced significant cybersecurity challenges, particularly in detecting sophisticated software supply chain attacks targeting ecosystems like the Python Package Index (PyPI) [1, 2]. PyPI hosts over 620,000 packages and facilitates billions of daily downloads, underscoring its central role in modern software development [3, 4]. However, its open nature and rapid scalability have made it a prime target for adversaries [5]. As of July 2024, 1.2% of total PyPI packages have been identified as malicious, highlighting growing security concerns [6–9]. These attacks are becoming more common as multi-stage threats that exploit vulnerabilities in the OSS ecosystem, including typosquatting, malware execution, remote access, and dynamic payload generation [10–12]. Such threats compromise the core security principles of confidentiality, integrity, and availability [2, 13]. Existing defense mechanisms, such as host-based firewalls and signature or rule-based malware scanners, struggle to counter these threats due to their inability to adapt to evolving, multi-stage adversarial tactics and their limited capacity for granular behavioral code analysis [14–16]. As a result, Malicious Detection Systems (MDS) have emerged as critical safeguards for the OSS ecosystem, using Machine Learning (ML) or Deep Learning (DL) models to identify and mitigate next-gen attacks [17–19].

---

*Corresponding author: `tanzir.mehedi@hdr.qut.edu.au`

39th Conference on Neural Information Processing Systems (NeurIPS 2025) Track on Datasets and Benchmarks.

The effectiveness of MDSs relies on their detection performance metrics, which require evaluation against comprehensive datasets containing both benign and malicious package behaviors [17, 5]. Widely adopted metadata and static dataset benchmarks, such as the PyPI Malware Registry [20], Backstabber's Knife Collection [21], DataDog [22], and PyPIGuard [23], have served as standards for MDS validation. However, recent studies highlight critical limitations in these datasets [5, 24]. For instance, metadata datasets capture only package details like descriptions and author profiles, while static datasets focus on code attributes such as function signatures and import statements, without executing packages during installation or post-installation [17, 5]. This omission severely limits their ability to detect dynamic threats such as typosquatting, remote access activation, and install-time-specific payload generation [14–16]. Hybrid datasets, which combine static and metadata features, partially address some threats but still fail to detect most of these threats, as they lack visibility into complex behaviors that occur during install-time and post-install-time [19, 24]. These limitations in existing datasets undermine the reliability of performance metrics, raising concerns about the generalizability of MDS evaluations to next-gen OSS supply chain attacks.

To address these challenges, this study introduces the `QUT-DV25` dataset, specifically designed to facilitate dynamic analysis of next-gen OSS software supply chain attacks within the PyPI ecosystem. `QUT-DV25` captures behavioral traces from 14,271 Python packages, 7,127 of which exhibit malicious behavior, through install and post-install-time in a controlled, isolated sandbox environment using an extended Berkeley Packet Filter (eBPF) kernel and user-level probes. A subset of high-value probes is selected from an initial pool of over 105 eBPF-based probes, focusing on those most effective for capturing behaviors relevant to malware execution. This tool enables real-time tracing without kernel modification and supports flexible, programmable monitoring in C or Python [25]. The dataset records 36 real-time features for each package, including system calls, network traffic, resource consumption, directory access patterns, dependency resolution, and installation behaviors. These features enable comprehensive analysis of dynamic attack vectors such as multiphase malware execution, remote access activation, and post-install-time-specific payload generation. A detailed characterization of the dataset's structure, threat coverage, and example is also provided. Furthermore, the performance of MDSs is evaluated based on this dataset as a binary classification task using multiple supervised ML and DL models. By bridging the gap between static and dynamic analysis, `QUT-DV25` empowers cybersecurity researchers to develop robust defenses against evolving OSS supply chain threats. The dataset[2], source code, and execution instructions [3] are publicly available, ensuring reproducibility and enabling further research. The key contributions of this study include:

- A controlled, isolated testbed framework to generate and collect datasets by installing and executing packages and capturing behaviors such as typosquatting, dynamic payload generation, and multiphase malware execution.

- `QUT-DV25`, a dataset of 14,271 packages, including 7,127 malicious ones, with 36 features across six categories that capture install-time and post-install-time behaviors previously unexplored for malicious package detection.

- A first-hand evaluation of four popular machine learning and two deep learning models on the proposed dataset is also provided as a baseline for further research.

This study is organized as follows: Section 2 reviews existing datasets. Section 3 outlines the dataset construction and details of the dataset. Section 4 presents technical validation and benchmarks. Section 5 discusses limitations and usage examples, and Section 6 addresses safety and ethical considerations. Finally, Section 7 concludes the study and outlines future directions.

## 2 Existing Datasets

The effectiveness of MDS datasets depends on two factors: coverage of modern threats and diversity of benign behaviors to reduce false positives [5]. Datasets lacking realistic adversarial contexts risk misleading evaluations [24]. Existing benchmarks are categorized as metadata, static, hybrid, and dynamic, each with limitations in modeling next-gen multi-stage attacks.

**Metadata datasets:** Metadata datasets focus on static, non-execution-based features such as package names, descriptions, version histories, and author profiles [17, 26, 27]. These datasets are widely adopted in MDSs due to their computational efficiency and ease of analysis, enabling rapid screening

---

[2]Dataset: `https://doi.org/10.7910/DVN/LBMXJY`; Package List: `https://qut-dv25.dysec.io`
[3]Code URL: `https://github.com/tanzirmehedi/QUT-DV25`

of large package repositories [17]. For example, Guo et al. [20] proposed the PyPI Malware Registry dataset, and Marc et al. [21] proposed the Backstabber's Knife Collection dataset to flag suspicious packages based on anomalies such as mismatched author credentials or irregular update patterns. However, metadata datasets fail to capture dynamic behaviors-such as system interactions during install-time or post-install-time-that are critical for detecting sophisticated threats like typosquatting, multiphase malware execution, and dynamic payload generation [14–16]. Attackers can exploit this limitation by crafting packages with plausible metadata while embedding malicious logic that activates post-deployment [24]. Consequently, MDSs relying solely on metadata suffer from high false positive rates, as benign packages with irregular metadata are misclassified, and malicious ones evade detection through metadata obfuscation [18].

**Static datasets:** Static datasets analyze code attributes such as function signatures, import statements, and control flow structures to identify malicious patterns without executing packages [5, 18, 28]. Datasets such as DataDog [22], developed by Datadog Security Labs, detect known threats-including hardcoded backdoors and command-and-control (C2) functionalities, matching code artifacts against curated malicious signature patterns. These datasets excel at identifying obvious malicious code and are computationally lightweight, making them scalable for large-scale repository scans [18, 28]. Static analysis, however, cannot detect install-time and post-install-time-specific threats such as multi-stage malware or environment-triggered malicious behavior [5]. For instance, a package with innocuous static code may execute a hidden script during installation to exfiltrate sensitive data scenario invisible to static inspection [18, 29, 14]. Additionally, techniques like code obfuscation or encryption easily bypass static detection, as they mask malicious intent until installation [30].

**Hybrid datasets:** Hybrid datasets integrate metadata and static code features to balance efficiency and depth, aiming to detect threats that evade single-mode analysis [19, 24]. For example, Samaana et al. [5] developed a hybrid dataset by combining metadata attributes with static code features to improve malicious package detection. Similarly, the PyPIGuard dataset, proposed by Iqbal et al. [23], combines package metadata with static code attributes to flag packages that may appear benign in isolation but exhibit suspicious patterns when analyzed holistically. This dataset improves detection of contextual threats, such as dependency confusion attacks, where malicious packages mimic legitimate names but contain altered code [24]. Despite their advantages, hybrid datasets remain limited by their lack of install-time behavioral data. For instance, they cannot model indirect dependency hijacking or post-deployment behaviors [14]. Advanced threats like polymorphic malware, which alters its code or behavior based on environmental cues, further evade hybrid detection due to the absence of dynamic execution traces [31]. These gaps undermine hybrid datasets' ability to address multi-stage attacks, where malicious activity unfolds progressively across install-time and post-install-time phases. Table 1 presents the existing datasets for detecting malicious packages in PyPI.

**eBPF-based dynamic datasets:** eBPF offers a powerful framework for real-time system monitoring, providing fine-grained visibility into install-time and post-install-time behaviors with low over-head [32, 25]. This capability has been used in security applications, such as ransomware detection through system call trace analysis and network anomaly identification [33, 34]. However, existing eBPF-based implementations predominantly rely on rule-based threat detection methodologies. For instance, Higuchi and Kobayashi [35] developed a ransomware detection system using eBPF-traced system call patterns, while Zhuravchak and Dudykevych [33] employed predefined behavioral rules for real-time ransomware analysis. Such rule-based approaches, though effective for known threats, lack adaptability to zero-day attacks due to their dependence on static signatures.

Table 1: Existing datasets for PyPI malicious package detection.

| Datasets | Detect manipulate metadata | Detect encoding technique | Dynamic payload generation | Detect typo-squatting | Remote access activation | Detect indirect dependencies |
|---|---|---|---|---|---|---|
| Metadata [20] | ◕ | ○ | ○ | ◕ | ○ | ○ |
| Static [22] | ● | ◕ | ○ | ◕ | ◕ | ○ |
| Hybrid [23] | ● | ◕ | ○ | ◕ | ◕ | ○ |
| QUT-DV25 | ● | ● | ● | ● | ● | ● |

**Note:** Malicious package detection - ● possible, ◕ partially possible, ○ not possible.

To enhance flexibility, tools like `bpftrace`, `bpftool`, and `bcc-tools` extend eBPF's utility by enabling dynamic tracing of low-level kernel and user-space events without kernel modifications [32]. These tools support programmable tracing in C or Python, facilitating the extraction of behavioral

signals such as system calls, network traffic, resource consumption, and directory access patterns [25]. While these traces provide a foundation for behavioral analysis, current ML- or DL-based MDSs often lack datasets that capture such dynamic, real-time system-level behaviors.

## 3 QUT-DV25 Dataset Construction

In this section, we first discuss the testbed configuration, followed by the dataset collection methodology and an overview of the dataset, including feature sets.

### 3.1 Testbed Configuration

The experimental testbed setup involves 16 Raspberry Pi devices $\mathcal{D}_{\mathrm{RPi}} = \{d_k\}_{k=1}^{16}$, each running Ubuntu 24.4 LTS with Python 3.8-3.12 in isolated virtualized environments. A private network $\mathcal{N}_{\mathrm{priv}} = \{\mathrm{Router}, \mathrm{Switch}, \mathrm{Raspberry\ Pi}\}$ ensures secure traffic flow. Behavioral monitoring is implemented using eBPF integrated into the Linux kernel $\mathcal{K} = \mathrm{v6.8.0\text{-}1012\text{-}raspi}$, with real-time tracing tools $\mathcal{T}_{\mathrm{bcc}} = \{\mathrm{bcc\text{-}tools}, \mathrm{bpftool}, \mathrm{bpftrace}\}$. Figure 1 shows a visualization of the isolated testbed configuration. To validate and scale the resulting dataset for ML and DL models, a high-performance computing cluster $\mathcal{C}_{\mathrm{HPC}} = \{c_k\}_{k=1}^m$ is employed, where each node features 16-core CPUs, NVIDIA A100 GPUs, and 128 GB of RAM.

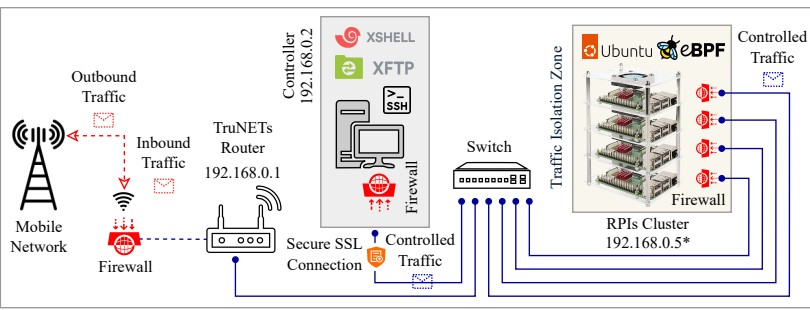

Figure 1: The isolated testbed configuration visualization for `QUT-DV25`.

### 3.2 Collection Methodology

We propose the `QUT-DV25` Dataset Framework, a structured methodology for constructing a dataset that captures both install-time and post-install-time behaviors of software packages. This framework is designed to meet the growing need for dynamic datasets in detecting multi-stage, next-gen software supply chain attacks, particularly within ecosystems such as PyPI. The framework systematically integrates three phases: (i) dataset collection, (ii) labeling and validation, and (iii) trace extraction, as illustrated in Figure 2.

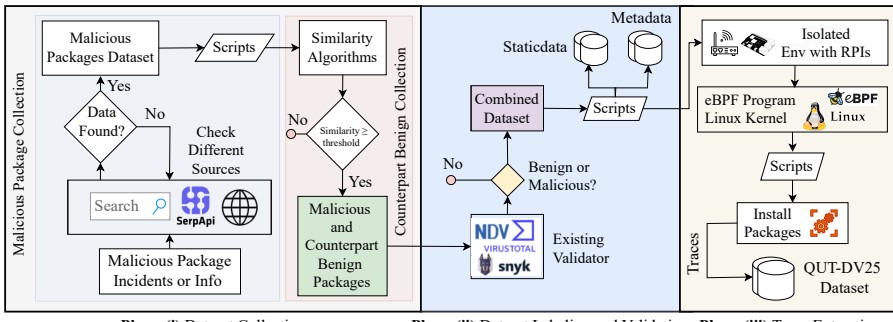

Figure 2: The overall framework for collecting the `QUT-DV25` dataset.

**Dataset collection:** In the absence of a centralized repository of malicious PyPI packages, we collect data from multiple threat intelligence sources denoted by $\mathcal{S} = \{S_1, \ldots, S_K\}$, where each $S_k$ corresponds to a source such as GitHub advisories or malware databases, and $K$ is the total number of such sources. The combined set of malicious packages is defined as $\mathcal{M} = \bigcup_{k=1}^K S_k = \{(n_m^i, v_m^i)\}_{i=1}^N$, where $n_m^i$ and $v_m^i$ denote the name and version of the $i$-th malicious package and $N$ is the total number of malicious samples collected. To enable comparative analysis and

support downstream classification tasks, we extract benign counterparts by defining the universe of benign packages as $\mathcal{U} = \{(n_b^j, v_b^j)\}_{j=1}^M$, where $n_b^j$ and $v_b^j$ denote the name and version of the $j$-th benign package, and $M$ is the total number of benign packages. We apply similarity algorithms $\text{sim}(n_m^i, n_b^j) \in [0, 1]$ to compute name-based similarity between malicious and benign packages. For each malicious package, we form a candidate set $\mathcal{C}^i = \{(n_b^j, v_b^j) \in \mathcal{U} \mid \text{sim}(n_m^i, n_b^j) \geq \tau\}$, where $\tau \in [0, 1]$ is a similarity threshold. If $\mathcal{C}^i \neq \emptyset$, we select the most similar benign package $(n_b^{*i}, v_b^{*i}) = \arg\max_{(n_b, v_b) \in \mathcal{C}^i} \text{sim}(n_m^i, n_b)$ and fetch the release date $r_b^{*i}$. The final similarity score is recorded as $s^{*i} = \text{sim}(n_m^i, n_b^{*i})$. Then, construct the final malicious-benign dataset $\mathcal{D}_{\text{mb}} = \{(n_m^i, v_m^i, n_b^{*i}, v_b^{*i}, r_b^{*i}, s^{*i}) \mid s^{*i} \geq \tau\}$ which serves as input for validation and dynamic trace extraction phases. Algorithm 1 outlines the process for retrieving counterpart benign packages.

---

**Algorithm 1:** Lexical Similarity-Based Benign Package Retrieval

---

**Input:** Malicious set $\mathcal{M} = \{(n_m^i, v_m^i)\}_{i=1}^N$; benign universe $\mathcal{U} = \{(n_b^j, v_b^j)\}_{j=1}^M$; threshold $\tau \in [0, 1]$
**Output:** Malicious-benign dataset $\mathcal{D}_{\text{mb}} = \{(n_m^i, v_m^i, n_b^{*i}, v_b^{*i}, r_b^{*i}, s^{*i})\}$
1 **Precondition:** Similarity metric; PyPI API accessible
2 **foreach** $(n_m^i, v_m^i) \in \mathcal{M}$ **do**
3  $\quad \mathcal{C}^i \leftarrow \{(n_b^j, v_b^j, s_{ij}) \mid (n_b^j, v_b^j) \in \mathcal{U}, \ s_{ij} = \text{sim}(n_m^i, n_b^j) \geq \tau\}$
4  $\quad$ **if** $\mathcal{C}^i \neq \emptyset$ **then**
5   $\quad\quad (n_b^{*i}, v_b^{*i}, s^{*i}) \leftarrow \arg\max \mathcal{C}^i$
6   $\quad\quad r_b^{*i} \leftarrow \texttt{QueryPyPI}(n_b^{*i})$
7   $\quad\quad \mathcal{D}_{\text{mb}} \leftarrow \mathcal{D}_{\text{mb}} \cup \{(n_m^i, v_m^i, n_b^{*i}, v_b^{*i}, r_b^{*i}, s^{*i})\}$
8  $\quad$ **end**
9 **end**
10 Export $\mathcal{D}_{\text{mb}}$ to file
11 **Postcondition:** Only pairs with $s^{*i} \geq \tau$ retained

---

**Dataset labeling and validation:** For each package $(n, v) \in \mathcal{D}_{\text{mb}}$, where $n$ and $v$ denote the package name and version respectively, a set of external threat intelligence validators $\mathcal{V} = \{\text{VirusTotal}, \text{NDV}, \text{Snyk}\}$ is queried. Each validator returns a label $\text{label}_k(n, v) \in \{0, 1, \bot\}$, indicating whether the package is malicious (1), benign (0), or inconclusive ($\bot$). The final label is determined as follows: $\text{Label}(n, v) = 1$ (malicious) if at least two validators return 1, i.e., $\sum_{k \in \mathcal{V}} \mathbb{I}[\text{label}_k(n, v) = 1] \geq 2$; $\text{Label}(n, v) = 0$ (benign) only if all validators agree the package is benign, i.e., $\sum_{k \in \mathcal{V}} \mathbb{I}[\text{label}_k(n, v) = 0] = |\mathcal{V}|$. If neither condition holds, due to inconclusive results, the label is assigned through manual inspection: $\text{Label}(n, v) = \text{ManualInspect}(n, v)$. The validated dataset is defined as $\mathcal{D}_{\text{valid}} = \{(n, v, \text{Label}(n, v)) \mid (n, v) \in \mathcal{D}_{\text{mb}}\}$.

In parallel, metadata features $\mathcal{X}(n, v)$ (e.g., author, version history, description) and static features $\mathcal{Y}(n, v)$ (e.g., import statements, function definitions) are extracted for each package. These are combined to construct the final labeled dataset $\mathcal{D}_{\text{final}} = \{(n, v, \text{Label}(n, v), \mathcal{X}(n, v), \mathcal{Y}(n, v)) \mid (n, v) \in \mathcal{D}_{\text{valid}}\}$. The dataset $\mathcal{D}_{\text{valid}}$ serves as the input to the next trace extraction step, while $\mathcal{D}_{\text{final}}$ is used as a benchmark for evaluating existing MDS models.

**Trace extraction:** The validated package set $\mathcal{D}_{\text{valid}} = \{(n_j, v_j, \text{Label}(n_j, v_j))\}_{j=1}^m$ serves as input for this phase. Each package archive is denoted by $\pi_j = (n_j, v_j) \in \mathcal{D}_{\text{valid}}$, and it is deployed to a uniformly random device $d_k = f(\pi_j)$ from the set of Raspberry Pi devices $\mathcal{D}_{\text{RPi}} = \{d_k\}_{k=1}^n$. Two binary indicators are defined: $\text{Deploy}(\pi_j, d_k)$ and $\text{Install}(\pi_j, d_k)$, which take the value 1 if the transfer and installation of $\pi_j$ on $d_k$ succeed, respectively. In cases where $\text{Install}(\pi_j, d_k) = 0$, partial installation of dependencies is occasionally observed. These cases form a subset $\mathcal{D}_{\text{partial}} \subset \mathcal{D}_{\text{valid}}$ and are of particular interest, as malicious payloads may persist through successfully installed subcomponents. Additionally, dependency resolution may implicitly introduce malicious variants: a benign package version $v_j$ of $n_j$ may cause the installation of a related version $v_j'$ such that $(n_j, v_j') \in \mathcal{D}_{\text{valid}}$ and $\text{Label}(n_j, v_j') = \text{Malicious}$. To account for such behavioral variability, these cases are retained for analysis.

During the install and post-install phases of each package, eBPF captures kernel- and user-space event sequences. However, at post-install-time, dormant malicious code can indefinitely delay execution, making open-ended runtime monitoring infeasible. To address this, we empirically tested 64 high-risk packages with observation windows from 60-600s and found no additional behavioral divergence

beyond 120s. Accordingly, we adopt a fixed post-installation tracing window of $\Delta = 120\,\text{s}$, which generates a sequence $\text{Trace}(\pi_j, d_k) \in \mathcal{S}^*$, where $\mathcal{S}^*$ denotes the space of trace sequences.

The trace extraction function $\text{Extract}(\pi_j, d_k)$ returns $\text{Trace}(\pi_j, d_k)$ only if both deployment and installation succeed, i.e., $\text{Deploy}(\pi_j, d_k) = \text{Install}(\pi_j, d_k) = 1$; otherwise, it outputs the empty set $\emptyset$. If $\text{Extract}(\pi_j, d_k) = \emptyset$, the environment on $d_k$ is reset and the process repeats with the same package until a valid trace is collected. The resulting valid trace is denoted $T_j = \text{Trace}(\pi_j, d_k)$, and the complete trace set is defined as $\mathcal{T} = \{T_j\}_{j=1}^m$. This trace set provides isolated and reproducible dynamic behavioral profiles for each package, supporting subsequent analysis (cf. Algorithm 2).

---

**Algorithm 2:** `QUT-DV25` Dynamic Trace Extraction

---

**Input:** Validated dataset $\mathcal{D}_{\text{valid}} = \{(n, v, \text{Label}(n, v))\}_{j=1}^m$; Raspberry Pi devices $\mathcal{D}_{\text{RPi}} = \{d_k\}_{k=1}^n$
**Output:** Traces $\mathcal{T} = \{T_j\}_{j=1}^m$

1 **Precondition:** Each $d_k$ runs an isolated Python 3.8–3.12 environment with eBPF support.
2 **Definitions:**
3      $f : \mathcal{D}_{\text{valid}} \to \mathcal{D}_{\text{RPi}}$     (uniform random device assignment)
4      $\text{Deploy}(\pi_j, d_k) = 1$ iff package $\pi_j$ is successfully transferred to $d_k$
5      $\text{Install}(\pi_j, d_k) = 1$ iff package $\pi_j$ installs successfully on $d_k$
6      $\text{Trace}(\pi_j, d_k) \in \mathcal{S}^*$ captures the eBPF event sequence during install-time and post-install-time (120s)
7      $\text{Extract}(\pi_j, d_k) = \text{Trace}(\pi_j, d_k)$ if $\text{Deploy}(\pi_j, d_k) = 1 \wedge \text{Install}(\pi_j, d_k) = 1$, else $\emptyset$
8 **for** $j \leftarrow 1$ **to** $m$ **do**
9      $d_k \leftarrow f(\pi_j)$
10      **if** $\text{Deploy}(\pi_j, d_k) = 0$ **or** $\text{Install}(\pi_j, d_k) = 0$ **then**
11          $T_j \leftarrow \emptyset$
12          **continue**
13      **end**
14      $T_j \leftarrow \text{Trace}(\pi_j, d_k)$
15 **end**
16 **return** $\mathcal{T} = \{T_j\}_{j=1}^m$

---

### 3.3 QUT-DV25 Data Records

This study analyzes a corpus of $|\mathcal{D}_{\text{valid}}| = 14,271$ Python packages, of which $7,127$ are labeled as malicious. Approximately 88% of these packages yielded successful installations, i.e., $\text{Install}(\pi_j, d_k) = 1$, while the remaining packages triggered direct install-time anomalies, such as system crashes, infinite loops, forced shutdowns, or authentication prompts-despite $\text{Install}(\pi_j, d_k) = 0$. To characterize behavioral variability across the dataset, a classification function $\text{Classify}(\pi_j) \in \mathcal{B} = \{\text{normal}, \text{compatibility}, \text{system}\}$ is introduced, mapping each package $\pi_j$ to a behavioral category based on its install-time and post-install-time outcomes in the isolated environment. Table 2 summarizes the characteristics of packages during install-time and post-install-time analysis.

Table 2: Characteristics of packages during install-time and post-install-time analysis.

| Install-time and post-install-time characteristics | Malicious | Benign |
|---|---|---|
| **Normal:** Successfully installed, metadata issues, setup/wheel issues | 6,864 | 6,905 |
| **Compatibility:** Mismatch, version issues, auth, naming, module issues | 236 | 202 |
| **System:** Freezing, infinity waiting, looping, shutdown, prerequisites | 27 | 37 |
| **Total:** | **7,127** | **7,144** |

**Feature sets and annotations:** To analyze install-time and post-install-time behaviors comprehensively, the system is instrumented using eBPF-based monitoring, which enables real-time capture of both kernel-space and user-space activity for each execution trace $T_j \in \mathcal{T}$. The feature set for each trace is denoted as $\mathcal{T}_j = \{F_j^{(i)}\}_{i=1}^q$, where each $F_j^{(i)}$ corresponds to a category of behavioral signals derived from eBPF probes. The observed traces are categorized into six primary trace types $\mathcal{F}$, mapping the raw trace $T_j$ into structured components $F_j^{(i)}$, each capturing a distinct dimension of install-time or post-install-time activity. Table 3 summarizes these eBPF-derived feature sets for each package $\pi_j$, describing their analytical focus and relevance to threat detection.

These features collectively define the vectorized representation $\mathcal{T}_j = \Phi(T_j) \in \mathbb{R}^q$, where $\Phi$ denotes the eBPF-based feature extraction operator and $q$ represents the dimensionality across all trace types. Unlike static or metadata-based approaches, this approach captures latent behaviors that manifest

only during install-time and post-install-time, allowing for the detection of advanced threats such as ransomware, backdoors, and privilege escalation. A detailed description of `QUT-DV25` feature types, along with representative examples, is provided in Appendix Table 7.

Table 3: Definitions of eBPF-based feature sets for package $\pi_j$.

| Feature Sets | Description |
|---|---|
| $F_j^{(ft)} = \text{FiletopTraces}(\pi_j)$ | File I/O process; detects abnormal file access or missing files. |
| $F_j^{(it)} = \text{InstallTraces}(\pi_j)$ | Dependency logs; indirect malicious installs. |
| $F_j^{(ot)} = \text{OpensnoopTraces}(\pi_j)$ | File open attempts; flags access to protected directories. |
| $F_j^{(tt)} = \text{TCPTraces}(\pi_j)$ | TCP flows; identifies connections to suspicious endpoints. |
| $F_j^{(st)} = \text{SysCallTraces}(\pi_j)$ | System call activity; indicates sabotage or privilege misuse. |
| $F_j^{(pt)} = \text{PatternTraces}(\pi_j)$ | Behavioral sequences; detects loops, or payload triggers. |

## 4 Technical Validation and Benchmarks of QUT-DV25

This section presents the technical validation and performance benchmarking of candidate ML and DL models for MDS, using the proposed `QUT-DV25` dataset.

**Data preparation:** The trace set $\mathcal{T} = \{T_j\}_{j=1}^m$ underwent preprocessing to ensure compatibility with ML and DL models. Duplicate packages were removed, incomplete traces discarded, and all installations were aligned to a uniform directory structure across devices to eliminate identifier bias. Each trace $T_j \in \mathcal{T}$ was transformed into a feature vector $x_j \in \mathbb{R}^d$, where categorical features were encoded as n-gram [36] frequency vectors $\phi_{\text{cat}}(T_j) \to \mathbb{R}^{d_c}$, and numerical features were scaled via min-max normalization: $x'_{j,i} = (x_{j,i} - \min(x_i))/(\max(x_i) - \min(x_i))$, for all $i$ in numerical features [37]. The final feature vector is $x_j = [\phi_{\text{cat}}(T_j), \phi_{\text{num}}(T_j)] \in \mathbb{R}^d$, suitable for training ML and DL models.

**Feature extraction and selection:** From the trace set $\mathcal{T}$, 62 candidate features were extracted, forming the set $CF = \{f_i\}_{i=1}^{62}$, where each $f_i$ denotes an attribute derived from the traces. To eliminate redundancy, features with a Pearson correlation coefficient [38] $|r_{ij}| > 0.50$ for any pair $(f_i, f_j) \in CF$ were pruned, resulting in an independent feature subset $IDF \subset CF$ with $|IDF| = 40$. For each feature $f \in IDF$, an importance score $IMS_m(f) \in [0,1]$ was computed using each model $m \in M_{ML} = \{\text{RF}, \text{DT}, \text{SVM}, \text{GB}\}$. This approach aligns conceptually with the feature-ranking mechanism [39]. The selected engineered feature set was defined as $SEF = \{f \in IDF \mid \max_{m \in M} IMS_m(f) > 0.08\}$, yielding $|SEF| = 36$, which corresponds to a 58% reduction in the original feature set. Also, to assess generalizability, H2O AutoML was used, and 32 of its top 36 features (89%) overlapped with our selected features, confirming the robustness of the feature selection approach. Subsequently, ML models were trained with the following hyperparameters: RF with `n_estimators=100` and `max_depth=8`; DT with `max_depth=8` and `min_samples_split=10`; SVM with a linear kernel; and GB with `n_estimators=100`, `max_depth=5`, and `learning_rate=0.1`. The dataset $\mathcal{D}$ was divided into training, validation, and testing subsets in the ratio $\mathcal{D}_{\text{train}} : \mathcal{D}_{\text{val}} : \mathcal{D}_{\text{test}} = 70{:}15{:}15$. Five-fold stratified cross-validation was employed for hyperparameter tuning, and final evaluation was conducted on $\mathcal{D}_{\text{test}}$. Furthermore, we evaluated the dataset using two widely used DL models, $m \in M_{DL} = \{\text{CNN}, \text{Transformer}\}$, configured with their standard baseline hyperparameter settings [40].

### 4.1 Experiments with ML Models

Each ML model $m \in M_{ML}$ was evaluated using accuracy $\mathcal{A}$, precision $\mathcal{P}$, recall $\mathcal{R}$, and F1-score $\mathcal{F}_1$, capturing overall detection correctness, robustness to false positives/negatives, and class imbalance.

The impact of trace-derived feature subsets $\mathcal{T} = \{T_j\}_{j=1}^m$ and their union `CombinedTraces` $= \cup \mathcal{T}$ on model performance was evaluated, as presented in Table 4. For each model $m \in M_{ML}$, features from `CombinedTraces` consistently yielded the highest performance (e.g., $\mathcal{A}_{\text{RF}} = 95.99\%$, $\mathcal{P}_{\text{RF}} = 96.00\%$, $\mathcal{F}_{1,\text{RF}} = 66.47\%$), outperforming any individual trace subset $T_j \in \mathcal{T}$. This improvement is attributed to feature complementarity: `FiletopTraces` captures resource I/O patterns; `OpensnoopTraces`, file access anomalies; `TCPTraces`, suspicious netflows; `SysCallTraces`, syscall anomalies; and `PatternTraces`, multi-stage attack sequences. Although `InstallTraces` alone show limited discriminative power ($\mathcal{A}_{\text{RF}} = 69.45\%$, $\mathcal{F}_{1,\text{RF}} = 66.47\%$) due to overlapping installation metadata, their inclusion in `CombinedTraces` enhanced attack coverage.

For standalone trace evaluation, performance varied across trace subsets $T_j \in \mathcal{T}$. `PatternTraces` demonstrated the highest effectiveness ($\mathcal{A}_{\text{RF}} = 94.62\%$, $\mathcal{F}_{1,\text{RF}} = 94.61\%$), reflecting its capacity to capture high-level behavioral signatures. `SysCallTraces` achieved strong performance ($\mathcal{A}_{\text{RF}} = 88.51\%$), while `FiletopTraces` and `TCPTraces` showed moderate results ($\mathcal{A}_{\text{RF}} = 92.01\%$ and $\mathcal{A}_{\text{RF}} = 83.74\%$, respectively). `InstallTraces` remained the least informative ($\mathcal{A}_{\text{RF}} = 69.45\%$), reinforcing their limited standalone utility. These findings highlight that combining heterogeneous trace types enables cross-domain behavioral reasoning and maximizes detection capability.

Table 4: Performance of ML models across features: bold indicates the best, ↑ second-best, ↓ third-best, and underline denotes the lowest value.

| | Metrics | Filetop | Install | Opensnoop | TCP | SysCall | Pattern | Combined |
|---|---|---|---|---|---|---|---|---|
| **RF** | $\mathcal{A}$ | 92.01 | 69.45 | 93.55 | 83.74 | 88.51 | 94.62 | **95.99** |
| | $\mathcal{P}$ | 92.10 | 80.28 | 93.62 | 83.74 | 88.51 | 94.95 | **96.00** |
| | $\mathcal{R}$ | 92.01 | 69.45 | 93.55 | 83.74 | 88.51 | 94.62 | **95.99** |
| | $\mathcal{F}_1$ | 92.00 | 66.47 | 93.55 | 83.74 | 88.51 | 94.61 | **96.02** |
| **DT** | $\mathcal{A}$ | 86.87 | 69.50 | 91.35 | 81.13 | 88.41 | 94.62 ↓ | 94.02 |
| | $\mathcal{P}$ | 86.87 | 80.84 | 91.36 | 81.21 | 88.41 | 94.95 ↓ | 94.36 |
| | $\mathcal{R}$ | 86.87 | 69.50 | 91.35 | 81.13 | 88.41 | 94.62 ↓ | 94.02 |
| | $\mathcal{F}_1$ | 86.87 | 66.43 | 91.35 | 81.11 | 88.41 | 94.61 ↓ | 94.28 |
| **SVM** | $\mathcal{A}$ | 89.77 | 68.65 | 80.05 | 80.47 | 85.56 | 94.53 | 95.28 ↑ |
| | $\mathcal{P}$ | 89.85 | 80.65 | 81.65 | 80.55 | 85.57 | 94.87 | 95.30 ↑ |
| | $\mathcal{R}$ | 89.77 | 68.65 | 80.05 | 80.47 | 85.56 | 94.53 | 95.28 ↑ |
| | $\mathcal{F}_1$ | 89.76 | 65.39 | 79.79 | 80.46 | 85.56 | 94.52 | 95.23 ↑ |
| **GB** | $\mathcal{A}$ | 87.38 | 67.16 | 91.54 | 80.47 | 85.61 | 94.58 | 94.11 |
| | $\mathcal{P}$ | 87.42 | 79.31 | 91.67 | 80.72 | 85.62 | 94.88 | 94.42 |
| | $\mathcal{R}$ | 87.38 | 67.16 | 91.54 | 80.47 | 85.61 | 94.58 | 94.61 |
| | $\mathcal{F}_1$ | 87.38 | 63.39 | 91.53 | 80.43 | 85.61 | 94.57 | 94.35 |

## 4.2 Experiments with DL Models

In addition to ML models, several DL models were also evaluated, $m \in M_{DL} = \{\text{CNN}, \text{Transformer}\}$, using the same performance metrics $\{\mathcal{A}, \mathcal{P}, \mathcal{R}, \mathcal{F}_1\}$. Each model $m \in M_{DL}$ was trained and validated on `CombinedTraces` for up to 200 epochs, denoted as $\mathcal{E} = \{e_i\}_{i=1}^{200}$. The CNN model achieved its optimal performance at $e^* = 72$, where the test accuracy and loss were $\mathcal{A}_{\text{CNN}}^{\text{test}} = 95.28\%$ and $\mathcal{L}_{\text{CNN}}^{\text{test}} = 0.2460$, respectively, as shown in Figures 3(a) and 3(b). Across epochs $e_i \in \mathcal{E}$, the evaluation metrics remained approximately stable, indicating limited performance gains from extended training or the use of more complex architectures.

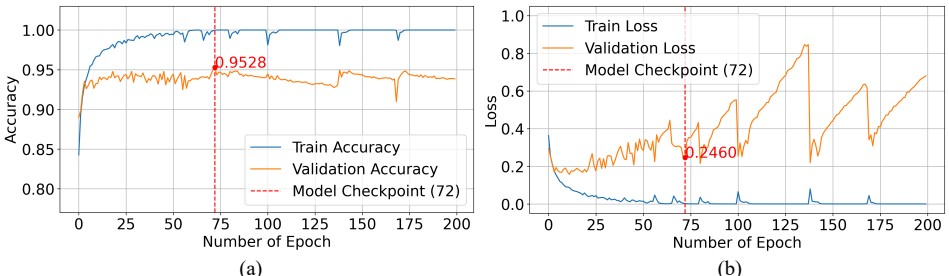

Figure 3: Training and validation (a) accuracies and (b) losses of the CNN model across epochs.

Furthermore, a comparative evaluation of all ML and DL models was conducted based on model complexity, computational efficiency, and deployability. As summarized in Table 5, each model $m \in \{M_{ML}, M_{DL}\}$ was assessed using performance metrics and timing parameters $\{t_{\text{train}}, t_{\text{val}}, t_{\text{test}}\}$. Although DL models achieved comparable accuracy ($\mathcal{A}_{DL} \approx \mathcal{A}_{ML}$), they required significantly higher computational cost ($t_{DL} \gg t_{ML}$), limiting real-time deployability. ML models offered a better trade-off between accuracy and efficiency ($\mathcal{A}_{ML} \simeq \mathcal{A}_{DL}, t_{ML} \ll t_{DL}$), and thus four ML models were selected as the primary benchmarks.

## 4.3 Comparison with Baseline Datasets

An effective MDS dataset must balance accuracy, efficiency, and generalization. The proposed `QUT-DV25` was compared against two ML-based baselines: (i) `MetadataDataset`, based on the

Table 5: Performance comparison of the selected ML and DL models; bold indicates the best values.

| | Selected Models | Accuracy (%) | Training Time (s) | Validation Time (s) | Test Time (s) |
|---|---|---|---|---|---|
| ML | RF | **95.99** | **0.4940** | 0.1512 | 0.1183 |
| | DT | 94.02 | 6.3058 | 0.0413 | 0.0388 |
| | SVM | 95.28 | 35.1509 | 0.4374 | 0.4151 |
| | GB | 94.11 | 46.9630 | **0.0277** | **0.0255** |
| DL | CNN | 95.28 | 5155.12 | 0.6954 | 0.5723 |
| | Transformer | 94.50 | 5532.74 | 0.7743 | 0.6505 |

method by Halder et al. [17], and (ii) `StaticDataset`, following the approach of Samaana et al. [5]. To ensure fairness, the original models, feature selection strategies, and hyperparameters were applied to features derived from the common corpus. As shown in Table 6, `QUT-DV25` with `CombinedTraces` and RF achieved the highest performance across all metrics: $\mathcal{A}_{RF} = 95.99\%$ and $\mathcal{F}_{1,RF} = 96.02\%$. Confusion matrix analysis further confirms its robustness with $TPR = 96.36\%$, $TNR = 98.26\%$, $FPR = 1.74\%$, and $FNR = 3.64\%$.

Table 6: Performance comparison with existing datasets; bold indicates the overall best values.

| Dataset | M | $\mathcal{A}$ (%) | $\mathcal{F}_1$ (%) | TPR (%) | TNR (%) | FPR (%) | FNR (%) |
|---|---|---|---|---|---|---|---|
| Metadata Dataset [20] | RF | 84.44 | 84.81 | 82.98 | 86.10 | 13.90 | 17.02 |
| | DT | 83.93 | 84.36 | 82.26 | 85.76 | 14.24 | 17.74 |
| | SVM | 80.47 | 81.60 | 77.26 | 84.59 | 15.41 | 22.74 |
| | GB | 83.46 | 84.25 | 80.52 | 87.04 | 12.96 | 19.48 |
| Static Dataset [22] | RF | 95.14 | 95.24 | 93.37 | 97.06 | 2.94 | 6.63 |
| | DT | 95.14 | 95.29 | 92.45 | 98.20 | 1.80 | 7.55 |
| | SVM | 95.32 | 95.30 | 96.01 | 94.65 | 5.35 | 3.99 |
| | GB | 94.90 | 95.08 | 92.06 | 98.19 | 1.81 | 7.94 |
| `QUT-DV25` | RF | **95.99** | **96.02** | 95.26 | 96.77 | 3.23 | 4.74 |
| | DT | 94.02 | 94.28 | 90.48 | **98.26** | **1.74** | 9.52 |
| | SVM | 95.28 | 95.23 | **96.36** | 94.24 | 5.76 | **3.64** |
| | GB | 94.11 | 94.35 | 90.71 | 98.16 | 1.84 | 9.29 |

In contrast, `MetadataDataset` exhibited high false positives (FPR $= 13.90\%$), attributable to its dependence on superficial package attributes, leading to poor generalization. Similarly, `StaticDataset` lacked install-time and post-install-time features, resulting in elevated false negatives (FNR $= 6.63\%$). Both baselines struggled with previously unseen samples. `QUT-DV25` outperforms both meta and static dataset baselines across $\mathcal{A}$, $\mathcal{F}_1$, and confusion matrix dimensions.

This dataset also enables the distinction between benign and malicious system call patterns [41]. By facilitating the differentiation of these patterns, it supports a more robust evaluation of their discriminative power in classification tasks. We also analyzed cases where ML models made mistakes. The RF model flagged 11 packages out of 1,000 recent PyPI packages that were originally labeled as benign. After review, six packages exhibited clear malicious behaviors such as data exfiltration, port scanning, and unauthorized remote access. These findings were reported to PyPI with supporting evidence from the `QUT-DV25` dataset, leading to the removal of four flagged packages (vermillion-0.5, eth-abcde-0.2.3, Pytonlib-0.0.0, and infoind-3897) from the platform. Two other flagged packages, PySocks-1.7.1 and escposprinter-6.2, are examples of dual-use tools. These packages can be used for both benign and malicious purposes. For instance, PySocks can hide network traffic, and escposprinter includes NetCat, which is often used to open backdoors. Although PyPI did not remove these packages and provided justification, the flagged behavior still raised important alerts. These are not simple classification mistakes; they highlight ambiguous, high-risk behaviors that require human judgment. Such cases demonstrate the limitations of binary labels in the real world. The remaining five packages were confirmed as FPs, but this number is small. These findings demonstrate the dataset's real-world effectiveness with low FNR and high precision.

## 5 Technical Limitations and Other Applications

**Technical limitations:** The performance of MDS depends on the quality and representativeness of the proposed dataset $\mathcal{D}$, where each sample comprises install-time and post-install-time traces $T_j \in \mathcal{T}$. These traces may include noise that obscures discriminative patterns. To mitigate this, all

$T_j$ were collected using eBPF within isolated Linux-based sandboxes, ensuring clean environments but also introducing package dependency issues, environment sensitivity, and setup overhead. To prevent inconsistencies due to package dependency reuse, each package was installed in a fresh virtual environment. However, the current setup may not capture environment-sensitive attacks (e.g., activate only on Windows or macOS) due to the lack of hardware and OS diversity. In particular, it may overlook actions triggered by user interaction, command-line arguments, or specific runtime conditions. Despite this limitation, the setup was deliberately chosen to prioritize safety, reproducibility, and cost-efficiency in managing the overhead of package execution. Moreover, since full support for eBPF is currently available only on Linux, our approach remains well representative of Linux-based systems, including widely used server and desktop distributions.

In addition, the extracted feature vectors $\phi(T_j) = x_j \in \mathbb{R}^d$ were high-dimensional, increasing dataset processing complexity. Dimensionality reduction techniques were applied to obtain a selected embedding $SEF_j \subset \mathbb{R}^d$, which preserves relevant semantics while improving efficiency. Since $\mathcal{D}$ is collected from PyPI packages, generalization to other ecosystems (e.g., NPM, Maven, RubyGems) may require retraining or domain adaptation. Furthermore, reliance on $\phi(T_j)$ may limit detection of delayed or obfuscated threats. To address this, future work will incorporate runtime traces and extend $\phi_{\text{post-install-time}}(T_j^{>120s})$ to enhance dataset detection robustness.

**Other applications of QUT-DV25:** The proposed dataset enables training ML models for malicious package detection using user and system-level traces and modeling multi-stage attacks such as dynamic payload execution and covert remote access. It also supports feature attribution studies for understanding behavioral indicators of compromise and provides a benchmark for evaluating dynamic detection systems. With eBPF-collected behavioral data from 14,271 PyPI packages, `QUT-DV25` offers a practical foundation for advancing dynamic malware analysis in software supply chains. This study benefits society by enhancing software supply chain security and leading to the removal of four previously undetected malicious PyPI packages. However, techniques like eBPF tracing, while safely handled in controlled environments here, could pose risks if misused for surveillance or exploitation.

## 6 Safety and Ethical Discussion

All benign packages used in this study were collected from publicly available Python packages in the PyPI repository. and all malicious packages collected from different publicly available websites based on a details security report. No user-generated or private data were included in the dataset $\mathcal{D}$, ensuring compliance with privacy norms and ethical research standards. The dynamic analysis was conducted in isolated, networked-controlled sandboxes to prevent accidental propagation of malicious behavior and ensure containment. The eBPF monitored only user and kernel-level behaviors $T_j \in \mathcal{T}$ within controlled environments, without logging personal or sensitive content. Also, the dataset $\mathcal{D}$ and feature extraction function $\phi(T_j)$ were designed solely for research purposes to advance open malware detection techniques. To discourage misuse, any release of $\mathcal{D}$ or $\phi(T_j)$ will undergo related institutional review and include documentation outlining ethical usage guidelines.

## 7 Conclusion and Future Works

Existing software supply chain security benchmarks fail to capture evolving threats such as typosquatting, delayed payloads, and covert remote access. To address this gap, `QUT-DV25` is introduced as a dynamic analysis dataset constructed in a controlled sandbox environment using eBPF kernel and user-level probes. The dataset models real-world PyPI packages by capturing 36 features, including system calls, network activity, and installation traces, across 14,271 packages, of which 7,127 exhibited malicious characteristics. Unlike static or metadata-based datasets that only represent surface-level attributes, `QUT-DV25` reflects modern attack behaviors observed during both install-time and post-install-time execution. Comparative evaluation demonstrates superior performance in modeling complex, dynamic threat vectors. `QUT-DV25` serves as a modern benchmark for dynamic malware detection and contributes to the advancement of next-gen software supply chain threat defenses. Future work includes extending the dataset to additional ecosystems, incorporating environment-sensitive attacks, and integrating ML frameworks for automated threat hunting across the open-source software supply chain ecosystem.

## 8 Acknowledgement

This research was supported by a QUT Postgraduate Research Award Scholarship and a QUT Tuition Fee Sponsorship. We acknowledge the support of QUT eResearch and Trusted Networks Lab for providing the computing and hardware facilities required to run our experiments.

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

# A Supplementary Material

Table 7: Detailed description of `QUT-DV-25` features with examples.

| Feature Name | Description | Examples |
|---|---|---|
| Package_Name | Package Name and Version | 1337z-4.4.7, 1337x-1.2.6 |
| *Filetop Traces* | | |
| Read_Processes | Processes in reading | pip reads setup.py for metadata |
| Write_Processes | Processes in writing | writes to site-packages and cached .whl |
| Read_Data_Transfer | Reading data transfer | pip reads .whl file from PyPI via HTTPS |
| Write_Data_Transfer | Writing data transfer | pip writes downloaded .whl into the local |
| File_Access_Processes | Processes in access files | Accesses _init_.py during installation |
| *Install Traces* | | |
| Total_Dependencies | Total number of dependencies | 2 (attrs-24.2.0; beautifulsoup4-0.1) |
| Direct_Dependencies | List of direct dependencies | 1 (beautifulsoup4-0.1) |
| Indirect_Dependencies | List of indirect dependencies | 1 (attrs-24.2.0) |
| *Opensnoop Traces* | | |
| Root_DIR_Access | Root directory access | 2 (/root/.ssh/authorized_keys) |
| Temp_DIR_Access | Temp directory access | 15 (/tmp/pip-wheel-pzrcqrtt/htaces.whl) |
| Home_DIR_Access | Home directory access | 55 (/home/Analysis/Env/1337z-4.4.7.) |
| User_DIR_Access | User directory access | 226 (/usr/lib/python3.12/lib-dynload) |
| Sys_DIR_Access | System directory access | 12 (/sys/kernel/net/ipv4/ip_forward) |
| Etc_DIR_Access | Etc directory access | 116 (/etc/host.conf, /etc/nftables.conf) |
| Other_DIR_Access | Access to other directories | 17 (/proc/sys/net/ipv4/conf, /.ssh) |
| *TCP Traces* | | |
| State_Transition | TCP lifecycle transitions | {CLOSE -> ->: 15, SYN_SENT} |
| Local_IPs_Access | Access to local IP addresses | 2 (192.168.0.51, 192.168.0.1) |
| Remote_IPs_Access | Access to remote IP addresses | 2 (151.101.0.223, 3.164.36.120) |
| Local_Port_Access | Access to local ports | 3 (52904, 53158, 34214) |
| Remote_Port_Access | Access to remote ports | 3 (443, 23, 6667) |
| *SysCall Traces* | | |
| IO_Operations | I/O operations performed | `ioctl, poll, readv` |
| File_Operations | File-related system calls | `open, openat, creat` |
| Network_Operations | Network-related operations | `socket, connect, accept` |
| Time_Operations | Time-based operations | `clock_gettime, timer_delete` |
| Security_Operations | Security-related sys calls | `getuid, setuid, setgid` |
| Process_Operations | Process management sys calls | `fork, vfork, clone` |
| *Pattern Traces* | | |
| Pattern_1 | File metadata retrieval | `newfstatat->openat->fstat` |
| Pattern_2 | Reading data from a file | `read->pread64->lseek` |
| Pattern_3 | Writing data to a file | `write->pwrite64->fsync` |
| Pattern_4 | Network socket creation | `socket->bind->listen` |
| Pattern_5 | Creating a new process | `fork->execve->wait4` |
| Pattern_6 | Memory mapping | `mmap->mprotect->munmap->no-fd` |
| Pattern_7 | File descriptor management | `dup->dup2->close->stdout` |
| Pattern_8 | Inter-process communication | `pipe->write->read->pipe-fd` |
| Pattern_9 | File locking | `fcntl->lockf->close->file-fd` |
| Pattern_10 | Error handling | `open->read->error=ENOENT->no-fd` |
| Labels | Classification level | [1,0] |

# NeurIPS Paper Checklist

1. **Claims**

   Question: Do the main claims made in the abstract and introduction accurately reflect the paper's contributions and scope?

   Answer: [Yes]

   Justification: The abstract and introduction clearly state the key contributions of our study, including the development of an isolated testbed framework, the creation of the QUT-DV25 dataset, and the baseline evaluation using multiple ML models. These claims align closely with the detailed descriptions and results presented in Sections 3 through 6, and we have made sure not to overstate the scope or generalizability of our findings.

   Guidelines:

   - The answer NA means that the abstract and introduction do not include the claims made in the paper.
   - The abstract and/or introduction should clearly state the claims made, including the contributions made in the paper and important assumptions and limitations. A No or NA answer to this question will not be perceived well by the reviewers.
   - The claims made should match theoretical and experimental results, and reflect how much the results can be expected to generalize to other settings.
   - It is fine to include aspirational goals as motivation as long as it is clear that these goals are not attained by the paper.

2. **Limitations**

   Question: Does the paper discuss the limitations of the work performed by the authors?

   Answer: [Yes]

   Justification: Section 5, 'Technical Limitations and Other Applications,' outlines several constraints of the study, including the platform dependency of our Linux-based testbed, potential noise in behavior traces, and the limited post-install-time observation window, which may overlook delayed or obfuscated threats. We also acknowledge that our dataset is specific to PyPI and may not generalize to other ecosystems without appropriate domain adaptation. The high dimensionality of extracted features required dimensionality reduction for practical model training. Additionally, each package was installed in isolation, which may not reflect real-world scenarios where dependencies interact. Although the approach was evaluated on seven datasets to provide broader empirical support, the scope of our claims is limited by these experimental conditions. Broader validation across multiple ecosystems and threat models is necessary to fully assess generalizability.

   Guidelines:

   - The answer NA means that the paper has no limitation while the answer No means that the paper has limitations, but those are not discussed in the paper.
   - The authors are encouraged to create a separate "Limitations" section in their paper.
   - The paper should point out any strong assumptions and how robust the results are to violations of these assumptions (e.g., independence assumptions, noiseless settings, model well-specification, asymptotic approximations only holding locally). The authors should reflect on how these assumptions might be violated in practice and what the implications would be.
   - The authors should reflect on the scope of the claims made, e.g., if the approach was only tested on a few datasets or with a few runs. In general, empirical results often depend on implicit assumptions, which should be articulated.
   - The authors should reflect on the factors that influence the performance of the approach. For example, a facial recognition algorithm may perform poorly when image resolution is low or images are taken in low lighting. Or a speech-to-text system might not be used reliably to provide closed captions for online lectures because it fails to handle technical jargon.
   - The authors should discuss the computational efficiency of the proposed algorithms and how they scale with dataset size.

- If applicable, the authors should discuss possible limitations of their approach to address problems of privacy and fairness.
- While the authors might fear that complete honesty about limitations might be used by reviewers as grounds for rejection, a worse outcome might be that reviewers discover limitations that aren't acknowledged in the paper. The authors should use their best judgment and recognize that individual actions in favor of transparency play an important role in developing norms that preserve the integrity of the community. Reviewers will be specifically instructed to not penalize honesty concerning limitations.

3. **Theory assumptions and proofs**

   Question: For each theoretical result, does the paper provide the full set of assumptions and a complete (and correct) proof?

   Answer: [NA]

   Justification: The study does not contain any theoretical results or proofs. This study focuses on dataset construction, benchmarking, and analysis of malicious behavior in software packages.

   Guidelines:

   - The answer NA means that the paper does not include theoretical results.
   - All the theorems, formulas, and proofs in the paper should be numbered and cross-referenced.
   - All assumptions should be clearly stated or referenced in the statement of any theorems.
   - The proofs can either appear in the main paper or the supplemental material, but if they appear in the supplemental material, the authors are encouraged to provide a short proof sketch to provide intuition.
   - Inversely, any informal proof provided in the core of the paper should be complemented by formal proofs provided in appendix or supplemental material.
   - Theorems and Lemmas that the proof relies upon should be properly referenced.

4. **Experimental result reproducibility**

   Question: Does the paper fully disclose all the information needed to reproduce the main experimental results of the paper to the extent that it affects the main claims and/or conclusions of the paper (regardless of whether the code and data are provided or not)?

   Answer: [Yes]

   Justification: The study provides detailed descriptions of the dataset construction process (Section 3), including the sandbox environment, data collection methodology, and feature extraction. Section 4 outlines the experimental setup, including preprocessing steps, the four ML models used, and their evaluation metrics. To further support reproducibility, we release the dataset with complete metadata and validation source codes. Additional details about the dataset are provided in the Appendix to facilitate deeper inspection and reuse.

   Guidelines:

   - The answer NA means that the paper does not include experiments.
   - If the paper includes experiments, a No answer to this question will not be perceived well by the reviewers: Making the paper reproducible is important, regardless of whether the code and data are provided or not.
   - If the contribution is a dataset and/or model, the authors should describe the steps taken to make their results reproducible or verifiable.
   - Depending on the contribution, reproducibility can be accomplished in various ways. For example, if the contribution is a novel architecture, describing the architecture fully might suffice, or if the contribution is a specific model and empirical evaluation, it may be necessary to either make it possible for others to replicate the model with the same dataset, or provide access to the model. In general. releasing code and data is often one good way to accomplish this, but reproducibility can also be provided via detailed instructions for how to replicate the results, access to a hosted model (e.g., in the case of a large language model), releasing of a model checkpoint, or other means that are appropriate to the research performed.

- While NeurIPS does not require releasing code, the conference does require all submissions to provide some reasonable avenue for reproducibility, which may depend on the nature of the contribution. For example
  (a) If the contribution is primarily a new algorithm, the paper should make it clear how to reproduce that algorithm.
  (b) If the contribution is primarily a new model architecture, the paper should describe the architecture clearly and fully.
  (c) If the contribution is a new model (e.g., a large language model), then there should either be a way to access this model for reproducing the results or a way to reproduce the model (e.g., with an open-source dataset or instructions for how to construct the dataset).
  (d) We recognize that reproducibility may be tricky in some cases, in which case authors are welcome to describe the particular way they provide for reproducibility. In the case of closed-source models, it may be that access to the model is limited in some way (e.g., to registered users), but it should be possible for other researchers to have some path to reproducing or verifying the results.

5. **Open access to data and code**

   Question: Does the paper provide open access to the data and code, with sufficient instructions to faithfully reproduce the main experimental results, as described in supplemental material?

   Answer: [Yes]

   Justification: We provide open access to both the dataset and source code, along with detailed instructions to replicate all experiments. The dataset is hosted at `https://doi.org/10.7910/DVN/LBMXJY` and includes a complete Croissant metadata file for standardized documentation and reproducibility. The technical validation source code is available at `https://github.com/tanzirmehedi/QUT-DV25`.

   Guidelines:
   - The answer NA means that paper does not include experiments requiring code.
   - Please see the NeurIPS code and data submission guidelines (`https://nips.cc/public/guides/CodeSubmissionPolicy`) for more details.
   - While we encourage the release of code and data, we understand that this might not be possible, so "No" is an acceptable answer. Papers cannot be rejected simply for not including code, unless this is central to the contribution (e.g., for a new open-source benchmark).
   - The instructions should contain the exact command and environment needed to run to reproduce the results. See the NeurIPS code and data submission guidelines (`https://nips.cc/public/guides/CodeSubmissionPolicy`) for more details.
   - The authors should provide instructions on data access and preparation, including how to access the raw data, preprocessed data, intermediate data, and generated data, etc.
   - The authors should provide scripts to reproduce all experimental results for the new proposed method and baselines. If only a subset of experiments are reproducible, they should state which ones are omitted from the script and why.
   - At submission time, to preserve anonymity, the authors should release anonymized versions (if applicable).
   - Providing as much information as possible in supplemental material (appended to the paper) is recommended, but including URLs to data and code is permitted.

6. **Experimental setting/details**

   Question: Does the paper specify all the training and test details (e.g., data splits, hyperparameters, how they were chosen, type of optimizer, etc.) necessary to understand the results?

   Answer: [Yes]

   Justification: The study provides comprehensive details of the experimental setup in Section 4, including dataset splits, preprocessing steps, model configurations, and training parameters for each baseline method. Technical validation implementation details and scripts are available in the GitHub repository (`https://github.com/tanzirmehedi/QUT-DV25`).

Guidelines:

- The answer NA means that the paper does not include experiments.
- The experimental setting should be presented in the core of the paper to a level of detail that is necessary to appreciate the results and make sense of them.
- The full details can be provided either with the code, in appendix, or as supplemental material.

7. **Experiment statistical significance**

Question: Does the paper report error bars suitably and correctly defined or other appropriate information about the statistical significance of the experiments?

Answer: [Yes]

Justification: The study reports standard ML performance metrics such as accuracy, precision, recall, and F1-score to evaluate the effectiveness of the proposed approach. However, it does not include error bars, confidence intervals, or statistical significance tests. Future work could incorporate such analysis to strengthen the statistical rigor of the findings.

Guidelines:

- The answer NA means that the paper does not include experiments.
- The authors should answer "Yes" if the results are accompanied by error bars, confidence intervals, or statistical significance tests, at least for the experiments that support the main claims of the paper.
- The factors of variability that the error bars are capturing should be clearly stated (for example, train/test split, initialization, random drawing of some parameter, or overall run with given experimental conditions).
- The method for calculating the error bars should be explained (closed form formula, call to a library function, bootstrap, etc.)
- The assumptions made should be given (e.g., Normally distributed errors).
- It should be clear whether the error bar is the standard deviation or the standard error of the mean.
- It is OK to report 1-sigma error bars, but one should state it. The authors should preferably report a 2-sigma error bar than state that they have a 96% CI, if the hypothesis of Normality of errors is not verified.
- For asymmetric distributions, the authors should be careful not to show in tables or figures symmetric error bars that would yield results that are out of range (e.g. negative error rates).
- If error bars are reported in tables or plots, The authors should explain in the text how they were calculated and reference the corresponding figures or tables in the text.

8. **Experiments compute resources**

Question: For each experiment, does the paper provide sufficient information on the computer resources (type of compute workers, memory, time of execution) needed to reproduce the experiments?

Answer: [Yes]

Justification: The study describes the computational environment used for dataset generation and model evaluation, including the use of isolated Linux-based sandboxes, virtual environments, and a machine with 32-core CPUs and 128GB RAM. Execution times for key steps such as dynamic tracing and feature extraction are summarized in Section 4.1, and the GitHub repository code includes runtime details of each algorithm.

Guidelines:

- The answer NA means that the paper does not include experiments.
- The paper should indicate the type of compute workers CPU or GPU, internal cluster, or cloud provider, including relevant memory and storage.
- The paper should provide the amount of compute required for each of the individual experimental runs as well as estimate the total compute.

- The paper should disclose whether the full research project required more compute than the experiments reported in the paper (e.g., preliminary or failed experiments that didn't make it into the paper).

9. **Code of ethics**

    Question: Does the research conducted in the paper conform, in every respect, with the NeurIPS Code of Ethics https://neurips.cc/public/EthicsGuidelines?

    Answer: [Yes]

    Justification: The research complies fully with the NeurIPS Code of Ethics. The dataset was collected from publicly available Python packages on PyPI, and all experiments were conducted in isolated environments to prevent harm. This ensured that no computing resources were damaged or disrupted due to malicious packages, and that the broader institutional network remained unaffected. No personal or sensitive data was used, and no users were affected. Ethical risks, such as potential misuse, were considered and discussed in Section 6. The release of the dataset and code follows open science best practices, including transparency and reproducibility.

    Guidelines:

    - The answer NA means that the authors have not reviewed the NeurIPS Code of Ethics.
    - If the authors answer No, they should explain the special circumstances that require a deviation from the Code of Ethics.
    - The authors should make sure to preserve anonymity (e.g., if there is a special consideration due to laws or regulations in their jurisdiction).

10. **Broader impacts**

    Question: Does the paper discuss both potential positive societal impacts and negative societal impacts of the work performed?

    Answer: [Yes]

    Justification: The study has a significant positive societal impact by advancing the detection of malicious software in open-source ecosystems. ML analysis using the dataset identified four malicious PyPI packages-previously labeled benign and with thousands of downloads-which were reported and subsequently removed, showcasing the practical security utility. However, the techniques, like eBPF tracing, could be misused for invasive monitoring or surveillance if applied inappropriately. The potential positive and negative impacts of the dataset are addressed in the final sentences of Section 5 of the study.

    Guidelines:

    - The answer NA means that there is no societal impact of the work performed.
    - If the authors answer NA or No, they should explain why their work has no societal impact or why the paper does not address societal impact.
    - Examples of negative societal impacts include potential malicious or unintended uses (e.g., disinformation, generating fake profiles, surveillance), fairness considerations (e.g., deployment of technologies that could make decisions that unfairly impact specific groups), privacy considerations, and security considerations.
    - The conference expects that many papers will be foundational research and not tied to particular applications, let alone deployments. However, if there is a direct path to any negative applications, the authors should point it out. For example, it is legitimate to point out that an improvement in the quality of generative models could be used to generate deepfakes for disinformation. On the other hand, it is not needed to point out that a generic algorithm for optimizing neural networks could enable people to train models that generate Deepfakes faster.
    - The authors should consider possible harms that could arise when the technology is being used as intended and functioning correctly, harms that could arise when the technology is being used as intended but gives incorrect results, and harms following from (intentional or unintentional) misuse of the technology.
    - If there are negative societal impacts, the authors could also discuss possible mitigation strategies (e.g., gated release of models, providing defenses in addition to attacks, mechanisms for monitoring misuse, mechanisms to monitor how a system learns from feedback over time, improving the efficiency and accessibility of ML).

11. **Safeguards**

Question: Does the paper describe safeguards that have been put in place for responsible release of data or models that have a high risk for misuse (e.g., pretrained language models, image generators, or scraped datasets)?

Answer: [Yes]

Justification: The dataset and feature extraction function were specifically designed for research purposes to advance open malware detection techniques. To prevent misuse, any release of these resources will undergo an institutional review and be accompanied by documentation outlining ethical usage guidelines. The research adheres to privacy norms, ensuring that no user-generated or private data were included. All testing and data collection were conducted in isolated, controlled environments, not connected to the broader institutional network or devices, to mitigate any unintended misuse and to safeguard organizational infrastructure.

Guidelines:

- The answer NA means that the paper poses no such risks.
- Released models that have a high risk for misuse or dual-use should be released with necessary safeguards to allow for controlled use of the model, for example by requiring that users adhere to usage guidelines or restrictions to access the model or implementing safety filters.
- Datasets that have been scraped from the Internet could pose safety risks. The authors should describe how they avoided releasing unsafe images.
- We recognize that providing effective safeguards is challenging, and many papers do not require this, but we encourage authors to take this into account and make a best faith effort.

12. **Licenses for existing assets**

Question: Are the creators or original owners of assets (e.g., code, data, models), used in the paper, properly credited and are the license and terms of use explicitly mentioned and properly respected?

Answer: [Yes]

Justification: All assets used in this study, including the dataset and code, are properly credited and the relevant licenses are respected. The dataset is publicly available under the Creative Commons CC0 1.0 Universal Public Domain Dedication (CC0 1.0). The code repository is hosted on GitHub. The terms of use and licenses for all external assets are clearly stated and adhered to in the study. Additionally, proper citation of the original sources is provided.

Guidelines:

- The answer NA means that the paper does not use existing assets.
- The authors should cite the original paper that produced the code package or dataset.
- The authors should state which version of the asset is used and, if possible, include a URL.
- The name of the license (e.g., CC-BY 4.0) should be included for each asset.
- For scraped data from a particular source (e.g., website), the copyright and terms of service of that source should be provided.
- If assets are released, the license, copyright information, and terms of use in the package should be provided. For popular datasets, `paperswithcode.com/datasets` has curated licenses for some datasets. Their licensing guide can help determine the license of a dataset.
- For existing datasets that are re-packaged, both the original license and the license of the derived asset (if it has changed) should be provided.
- If this information is not available online, the authors are encouraged to reach out to the asset's creators.

13. **New assets**

Question: Are new assets introduced in the paper well documented and is the documentation provided alongside the assets?

Answer: [Yes]

Justification: The new asset introduced in this study, the QUT-DV25 dataset, is well documented both on GitHub and Dataverse. Comprehensive documentation regarding the dataset's structure, usage, and licensing is provided alongside the dataset on both platforms. The dataset is publicly available under the Creative Commons CC0 1.0 Universal Public Domain Dedication (CC0 1.0), and all consent and ethical considerations were properly followed during its creation.

Guidelines:

- The answer NA means that the paper does not release new assets.
- Researchers should communicate the details of the dataset/code/model as part of their submissions via structured templates. This includes details about training, license, limitations, etc.
- The paper should discuss whether and how consent was obtained from people whose asset is used.
- At submission time, remember to anonymize your assets (if applicable). You can either create an anonymized URL or include an anonymized zip file.

14. **Crowdsourcing and research with human subjects**

Question: For crowdsourcing experiments and research with human subjects, does the paper include the full text of instructions given to participants and screenshots, if applicable, as well as details about compensation (if any)?

Answer: [NA] .

Justification: The study does not involve any crowdsourcing experiments or research with human subjects. All data used were sourced from publicly available software repositories and cybersecurity reports, with no interaction with or data collection from human participants.

Guidelines:

- The answer NA means that the paper does not involve crowdsourcing nor research with human subjects.
- Including this information in the supplemental material is fine, but if the main contribution of the paper involves human subjects, then as much detail as possible should be included in the main paper.
- According to the NeurIPS Code of Ethics, workers involved in data collection, curation, or other labor should be paid at least the minimum wage in the country of the data collector.

15. **Institutional review board (IRB) approvals or equivalent for research with human subjects**

Question: Does the paper describe potential risks incurred by study participants, whether such risks were disclosed to the subjects, and whether Institutional Review Board (IRB) approvals (or an equivalent approval/review based on the requirements of your country or institution) were obtained?

Answer: [NA]

Justification: The study does not involve any human subjects or participant data. All datasets were constructed using publicly available software packages and security reports, with no interaction or data collection from individuals, thus IRB approval was not applicable.

Guidelines:

- The answer NA means that the paper does not involve crowdsourcing nor research with human subjects.
- Depending on the country in which research is conducted, IRB approval (or equivalent) may be required for any human subjects research. If you obtained IRB approval, you should clearly state this in the paper.

- We recognize that the procedures for this may vary significantly between institutions and locations, and we expect authors to adhere to the NeurIPS Code of Ethics and the guidelines for their institution.
- For initial submissions, do not include any information that would break anonymity (if applicable), such as the institution conducting the review.

16. **Declaration of LLM usage**

    Question: Does the paper describe the usage of LLMs if it is an important, original, or non-standard component of the core methods in this research? Note that if the LLM is used only for writing, editing, or formatting purposes and does not impact the core methodology, scientific rigorousness, or originality of the research, declaration is not required.

    Answer: [NA]

    Justification: LLMs were used only for editing purposes such as grammar correction, spelling, and formatting. They were not involved in the development of core methods, data analysis, or scientific contributions of the paper.

    Guidelines:

    - The answer NA means that the core method development in this research does not involve LLMs as any important, original, or non-standard components.
    - Please refer to our LLM policy (`https://neurips.cc/Conferences/2025/LLM`) for what should or should not be described.

