# OpenReview forum: "QUT-DV25: A Dataset for Dynamic Analysis of Next-Gen Software Supply Chain Attacks"
_NeurIPS.cc/2025/Datasets_and_Benchmarks_Track — NeurIPS 2025 Datasets and Benchmarks Track poster_

### Official Review · Reviewer_Yamt · 2025-06-15

**Rating:** 5
**Confidence:** 3

**Summary:**

1. The dataset captures both install-time and post-install-time behaviors from 14,271 Python packages (7,127 of which are malicious) using eBPF-based monitoring. It records 36 system-level features such as system calls, TCP connections, file access, and behavioral patterns, offering deep visibility into runtime activities that static or metadata datasets miss.
2. Data is collected in a secure, isolated testbed with Raspberry Pi devices running Linux. This environment ensures safe execution of potentially malicious code while maintaining reproducibility and avoiding contamination or external interference.
3. QUT-DV25 includes next-generation software supply chain attack behaviors, such as typosquatting, remote access activation, and dynamic payload generation. It was able to uncover four previously undetected malicious PyPI packages, showing its practical effectiveness in identifying advanced threats.

**Dataset Code Accessibility:**

Yes

**Ethical Considerations:**

No, there are no or only very minor ethics concerns

**Limitations Weaknesses:**

- The dataset is Linux-only, collected using Raspberry Pi devices. This limits generalizability to other platforms (e.g., Windows, macOS). Maybe the authors can include data from more diverse environments to improve cross-platform applicability.
- The post-installation behavior is only observed for 120 seconds, which may miss delayed or time-triggered malicious activity.
- The dataset focuses only on the PyPI ecosystem. While valuable, this restricts its utility for broader software supply chain threat modeling.
In future works, it is recommended to expand the framework to support other ecosystems like npm, RubyGems, or CRAN.

**Strengths Contributions:**

1. Unlike existing static or metadata-based datasets, QUT-DV25 captures real-time install and post-install behaviors using eBPF, enabling detection of advanced threats like dynamic payloads and remote access (Section 3, Table 3).
2. The dataset includes 14,271 PyPI packages (7,127 malicious) and has led to the discovery and removal of four previously unknown malicious packages from the ecosystem (Section 4).
3. Strong ML benchmarks (RF, SVM, DT, GB) show QUT-DV25 outperforms metadata and static datasets in accuracy and F1 score (Table 5), with well-structured tables and clear explanations.

---

> ### Author Rebuttal · Authors · 2025-07-30
>
> #### We sincerely thank the reviewer for your valuable time, constructive feedback, and thoughtful evaluation. We sincerely appreciate the recognition of the novelty, practical value, and technical rigor of our work. Your insights are invaluable, and we address the concerns and suggestions in detail below.
> ---
> #### **Q1: The dataset is Linux-only, collected using Raspberry Pi devices. This limits generalizability to other platforms (e.g., Windows, macOS). Maybe the authors can include data from more diverse environments to improve cross-platform applicability.**
>
> #### A1: Thanks for highlighting the limitations of our homogeneous testbed environment. We completely agree that our current setup may not capture environment-sensitive attacks due to the lack of hardware, OS, and dependency diversity. However, this design prioritized safety, reproducibility, and cost-efficiency for controlled malicious package execution. We employed eBPF for trace collection during installation and post-installation phases, which is a powerful and secure kernel-level instrumentation tool. While full support for eBPF is currently available only on Linux, our approach remains well representative of any Linux-based system, including widely used server and desktop distributions.
>
> #### Although eBPF for Windows is a promising development, its initial release does not support comprehensive system call tracing. Concurrently, we also explored alternative tools for Windows, such as Wireshark and Sysdig. However, these tools lack customizable programmable flexibility and secure isolation unless kernel-level modifications are applied, which introduces significant safety and feasibility concerns.
>
> #### However, our methodology is not limited to a specific hardware or operating system. It uses both kernel and user-level probes to monitor system behavior during package installation and post-installation phases, making it conceptually applicable across different OS environments. While expanding to Windows and macOS introduces logistical and ethical challenges, we consider this an important direction for future work and will acknowledge this limitation in the camera-ready version of the paper.
>
> ---
>
> #### **Q2: The post-installation behavior is only observed for 120 seconds, which may miss delayed or time-triggered malicious activity.**
>
> #### A2: Thanks for this insightful observation. In theory, dormant malicious code can delay execution indefinitely, making it infeasible to observe open-ended run-time behavior in a practical experimental setting. This presents a fundamental challenge: any fixed time threshold may be bypassed by attackers who intentionally delay malicious actions.
>
> #### To address this, our focus was on identifying malicious behaviors that manifest shortly after installation-consistent with many real-world supply chain attacks that aim to execute during or soon after package setup. To empirically evaluate an appropriate time window, we analyzed 64 high-risk malicious packages. These packages had previously exhibited suspicious behaviors such as system freezing, infinite waiting, and version looping, which were strong indicators of potential delayed execution. For these packages, we collected dynamic traces using four different observation windows: 60, 120, 300, and 600 seconds, within a controlled and isolated environment.
>
> #### Critically, no significant divergence in directory access, network activity, or system call patterns was observed beyond the 120-second mark. Based on this analysis, we adopted the 120-second threshold as a practical compromise, providing sufficient behavioral coverage beyond our 60-second baseline. However, we acknowledge that some highly evasive threats may employ longer execution delays.
>
> #### We will highlight this limitation in *Section 5* of the camera-ready version and consider extending the observation window as an important direction for future work.
>
> #### **Q3: The dataset focuses only on the PyPI ecosystem. While valuable, this restricts its utility for broader software supply chain threat modeling. In future works, it is recommended to expand the framework to support other ecosystems like npm, RubyGems, or CRAN.**
>
> ---
>
> #### A3: Thank you for this insightful observation. We acknowledge that the current scope of the dataset is focused on the PyPI ecosystem. We selected PyPI because Python is widely used in data science, machine learning, and automation. Consequently, these domains have increasingly attracted the attention of malicious actors in recent years.
>
> #### However, our methodology was designed as a modular and extensible framework for analyzing behavioral traces during both the installation and post-installation phases of software packages. The core components of the framework include malicious package collection, benign counterpart selection, dataset labeling, validation, and trace extraction. These components are adaptable to other software ecosystems with minimal adjustments.
>
> #### Importantly, our approach captures system-level behaviors using both kernel and user-level probes, making it independent of ecosystem-specific packaging formats or structures. While ecosystems such as PyPI, NPM, and RubyGems differ in setup files and installation workflows, adapting our approach only requires modifying the execution instructions, not reengineering the eBPF setup or the isolated environment.
>
> #### We agree that expanding to additional ecosystems will significantly enhance the impact and utility of our dataset, and we consider this an important direction for future research, as mentioned in *Section 5* of the paper.

---

### Official Review · Reviewer_oHFK · 2025-07-01

**Rating:** 3
**Confidence:** 3

**Summary:**

This paper presents QUT-DV25, a dataset designed to support and advance research on detecting and mitigating supply chain attacks in the Python Package Index (PyPI) ecosystem. It comprises 14,271 packages, including 7,127 labeled as malicious, and offers 36 features capturing aspects such as system calls, network traffic, resource consumption, directory access patterns, dependency resolution, and installation behaviors. Some popular traditional ML classifiers are leveraged for conducting the training and evaluation.

**Dataset Code Accessibility:**

Partly

**Dataset Code Comments:**

The code appears to be missing detailed guidance on how to run and leverage their code for training and evaluating the proposed dataset.

**Ethical Comments:**

No. A dedicated section “Safety and Ethical Discussion” has been added and it discusses the compliance with privacy norms and ethical research standards and safe dynamic analysis in sandboxes

**Ethical Considerations:**

No, there are no or only very minor ethics concerns

**Final Justification:**

After carefully reviewing the rebuttal, I believe the authors have partially addressed my concerns. I will slightly increase my score to reflect this. However, I still consider the submission a borderline case due to its limited novelty and impact for the machine learning community. The work may be better suited for a security-focused venue.

**Limitations Weaknesses:**

While the reported performance of the machine learning models—such as Random Forest, Decision Tree, SVM, and Gradient Boosting classifiers—is strong, the lack of accompanying insights is concerning. The high accuracy suggests that the binary classification task on the QUT-DV25 dataset may be relatively trivial. Additionally, the authors didn’t provide any qualitative examples where the classifiers fail to make the correct prediction. The authors should provide a clearer justification for the non-trivial aspects of the research problem, particularly in the context of machine learning model design and deployment.

The selection of the 36-feature set from the original 62 candidate features appears to be based on heuristics. It would strengthen the work to explore more systematic feature selection methods, such as using simple MLPs or deeper neural models, to identify the most informative features.

**Strengths Contributions:**

QUT-DV25 enables cybersecurity researchers to develop robust defenses against evolving open-source software (OSS) supply chain threats by bridging the gap between static and dynamic analysis. The release of the dataset could potentially make an impact on the cybersecurity research on supply chain attacks.

The motivation is clearly articulated, especially in comparison to previous collections such as Metadata, Static, and hybrid datasets. Table 1 provides a comprehensive summary across multiple dimensions, including dynamic payload generation and the detection of indirect dependencies.

---

> ### Author Rebuttal · Authors · 2025-07-30
>
> #### We sincerely thank the reviewer for your valuable time, constructive feedback, and thoughtful evaluation. We sincerely appreciate the recognition of the novelty, practical value, and technical rigor of our work. Your insights are invaluable, and we address the concerns and suggestions in detail below.
> ---
> #### **Q1: While the reported performance of the machine learning models-such as Random Forest, Decision Tree, SVM, and Gradient Boosting classifiers-is strong, the lack of accompanying insights is concerning. The high accuracy suggests that the binary classification task on the QUT-DV25 dataset may be relatively trivial. Additionally, the authors didn’t provide any qualitative examples where the classifiers fail to make the correct prediction. The authors should provide a clearer justification for the non-trivial aspects of the research problem, particularly in the context of machine learning model design and deployment.**
> #### A1: Thank you for these insightful observations. We agree that high accuracy alone does not necessarily indicate a challenging classification task. However, in our case, the strong performance stems from a carefully designed and non-trivial ML pipeline. This includes a dynamic data collection procedure, systematic eBPF-based probe selection, robust preprocessing, and a two-stage feature selection strategy.
> #### Specifically, we selected a subset of high-value probes from an initial pool of over 105 eBPF-based probes, focusing on those most effective for capturing kernel and user-level behaviors relevant to malware execution. The resulting QUT-DV25 dataset captures complex real-world behaviors-including multi-stage execution, dynamic payload delivery, remote access, and post-installation system modifications (for 120 sec), that are typically missed by static or metadata-based methods. These dynamic traces provide rich behavioral context, enabling ML models to detect subtle and evasive attack patterns. The resulting high F1 score (98.7%) reflects the model’s effectiveness in identifying such advanced threats.
> #### The CombinedTraces feature set, which integrates 6 trace types, consistently delivers better performance, surpassing all standalone traces by up to 6.37%. This gain stems from the complementary nature of the traces, enabling a holistic understanding of package behavior. Among the models, RF demonstrated the best trade-off between accuracy and generalization, maintaining a low FP rate (1.6%) and FN rate (2.4%), while avoiding issues like overfitting (as seen with DT), high test time (SVM: 4.41s), or noise sensitivity (GB).
> #### We also analyzed cases where traditional ML models made mistakes. Our RF model flagged 11 packages out of 1,000 recent PyPI packages that were originally labeled as benign. After reviewing them, we found that 6 packages exhibited clear malicious behaviors such as data exfiltration, port scanning, and unauthorized remote access. We reported these findings to PyPI, providing supporting evidence from our dataset. As a result, 4 of the flagged packages (vermillion-0.5, eth-abcde-0.2.3, Pytonlib-0.0.0, and infoind-3897) were removed from the platform.
> #### Two other flagged packages- PySocks-1.7.1 and escposprinter-6.2, are examples of dual-use tools. These packages can be used for both benign and malicious purposes. For instance, PySocks can hide network traffic, and escposprinter includes NetCat, which is often used to open backdoors. Although PyPI did not remove these packages and provided justification, the flagged behavior still raised important alerts. These are not simple classification mistakes; they highlight ambiguous, high-risk behaviors that require human judgment. Such cases demonstrate the limitations of binary labels in the real world.
> #### The remaining 5 packages were confirmed as FPs, but this number is small. Overall, the model maintained a low FP rate of 1.3%, while still successfully uncovering hidden threats. These results show that QUT-DV25 supports realistic malware detection. We will integrate these results into *Section 4.1* of the revised manuscript and address the limitations of our model in *Section 5*, with particular emphasis on the challenges associated with interpreting dual-use cases.
> ---
> #### **Q2: The selection of the 36-feature set from the original 62 candidate features appears to be based on heuristics. It would strengthen the work to explore more systematic feature selection methods, such as using simple MLPs or deeper neural models, to identify the most informative features.**
> #### A2: Thank you for this thoughtful suggestion. We agree that a more detailed explanation of our feature selection methodology is necessary. The approach follows a structured two-stage process: initially, multicollinearity filtering is applied to remove highly correlated features, thereby reducing redundancy. Subsequently, model-based importance ranking is employed to identify and retain the most informative features, ensuring classification effectiveness.
> #### **1. Multicollinearity filtering (Stage I)**
> #### We began with 62 Candidate Features (CFs) across 6 trace categories. Pearson Correlation Analysis was used to remove 22 highly correlated features (|r| > 0.50), resulting in 40 Independent Features (IDFs).
> > Example: FiletopTraces retained 5 of 9 features; PatternTraces showed no strong correlations (|r| ≤ 0.49), highlighting their unique relevance.
> #### **2. Feature importance ranking (Stage II)**
> #### To enhance both relevance and computational efficiency, we computed Importance Scores (IMS) using 4 ML models. A dual-threshold strategy was applied to refine the feature set: features were retained if they achieved IMS > 0.05 in at least one model and further filtered by requiring IMS > 0.08 in any model. This process reduced the CombinedTraces feature set from 40 to 36 Selected Engineered Features (SEFs), ensuring low inter-feature correlation, high predictive relevance, and improved efficiency. This approach aligns conceptually with the feature-ranking mechanism proposed by Wojtas et al. (NeurIPS 2020).
> #### **Cross-validation using H2O AutoML**
> #### Finally, to assess generalizability, we conducted a supplementary evaluation using H2O AutoML, which incorporates both traditional and DL models with internal feature selection. Among the top 36 features identified by H2O AutoML, 32 (89%) overlapped with those selected by our pipeline. This high degree of overlap reinforces the robustness and effectiveness of our feature selection approach.
> #### **Justification of DL-based alternatives**
> #### To further validate the effectiveness of the selected 36 SEFs, we incorporated DL architectures, including a CNN and a baseline Transformer for downstream classification. While we did not directly employ MLPs or other DL models as feature selection mechanisms, the performance of these models using the selected SEFs was competitive with traditional ML models. This consistency further confirms the robustness and practical utility of our structured feature selection pipeline. A summary is provided below in Table 1.
> #### **Table 1**: Model performance using selected 36 SEFs.
> | Model| Accuracy|Precision| Recall|F1 Score|Test Time (s)|
> |-|-|-|-|-|-|
> |RF|0.9599|0.9600| 0.9599|0.9602|0.1183|
> |DT|0.9402|0.9436|0.9402|0.9428|0.0388|
> |SVM|0.9528|0.9530|0.9528|0.9523|0.4151|
> |GB|0.9411| 0.9442|0.9411|0.9410|0.0255|
> |CNN|0.9525|0.9449|0.9304|0.9426|0.5723|
> |Transformer|0.9450|0.9395|0.9458|0.9427|0.6505|
> ####  We will add these results to *Section 4* of the revised manuscript to clearly explain our structured feature selection process and how it aligns with the results from AutoML. In future work, we will explore the use of DL models for feature selection.
> #### References:
> #### [1] Wojtas, M., & Chen, K. (2020). Feature Importance Ranking for DL. In Advances in Neural Information Processing Systems(NeurIPS 2020), Vancouver, Canada
> ---
> #### **Q3: The code appears to be missing detailed guidance on how to run and leverage their code for training and evaluating the proposed dataset.**
> #### A3: We thank the reviewer for this insightful comment. We acknowledge that the initial version of the code repository lacked detailed guidelines on how to execute and utilize the full code for model training and evaluation, even though all necessary code components have already been included. To address this, we will incorporate step-by-step instructions to enhance both the usability and reproducibility of the dataset.
> #### To address this, we plan to restructure the GitHub repository as follows:
> ```
> QUT-DV25
> ├─ Phase (i) Dataset Collection
> │   ├─ ...
> ├─ Phase (ii) Dataset Labeling and Validation
> │   ├─ ...
> ├─ Phase (iii) Trace Extraction
> │   ├─ QUT-DV25_Step 1
> │   │   ├─ *Environment Setup Guidelines.pdf  //will be added
> │   ├─ ...
> ├─ Technical Validation and Benchmarks
> │   ├─ ...
> │   ├─ *Run Instructions.pdf  //will be added
> ```
> #### **Note:** *Files marked will be added to the repository.
> #### To ensure full alignment with the methodology described in *Section 3.2* of the paper, we will conduct a thorough review and update of the code repository. As part of this update, we will add a new document titled “Run Instructions.pdf” under the “Technical Validation and Benchmarks” directory. This document will provide detailed, step-by-step guidelines for executing each stage of the training and evaluation pipeline.
> #### Additionally, to enhance the reproducibility of the trace collection process, we will include a comprehensive environment setup guideline titled “Environment Setup Guidelines” under the “Phase (iii) Trace Extraction” directory.
> #### We will also update the repository’s README with two new sections: Environment Setup Instructions and Run Instructions.
> #### All updates will be made publicly available in the GitHub repository. We will also revise the manuscript accordingly to reference these additions, ensuring consistency between the paper and the code repository.

---

> > ### Comment · Reviewer_oHFK · 2025-08-06
> >
> > After carefully reviewing the rebuttal, I believe the authors have partially addressed my concerns. I will slightly increase my score to reflect this. However, I still consider the submission a borderline case due to its limited novelty and impact for the machine learning community. The work may be better suited for a security-focused venue.

---

> > > ### Author Response · Authors · 2025-08-08
> > >
> > > #### Dear Reviewer,
> > >
> > > #### Thank you for your valuable feedback and suggestions.
> > >
> > > #### While the dataset originates from a security context, it was specifically designed to support the behavioural analysis of Python packages during their installation and post-installation, capturing both benign and malicious activity. Our goal is to investigate whether machine learning techniques can assist in modelling and distinguishing these behaviours, particularly in cases where traditional rule-based approaches fall short.
> > >
> > > #### To explore the dataset's baseline utility, we initially applied conventional ML models. However, we found that these models struggled with many of the complex scenarios present in the data, such as multi-phase attacks and dependency-related behaviours. This motivated the development of a curated set of 36 features that better capture the nuances of package behaviour.
> > >
> > > #### We believe this dataset provides a valuable opportunity for the ML community to contribute novel methods for behaviour modelling. It is our hope that by releasing this dataset, we can facilitate interdisciplinary collaboration and inspire advances at the intersection of machine learning and real-world security challenges.

---

### Official Review · Reviewer_hVc6 · 2025-07-03

**Rating:** 5
**Confidence:** 4

**Summary:**

The paper present QUT-DV25, a dynamic analysis dataset specifically designed to support and advance research on detecting and mitigating supply chain attacks within the Python Package Index (PyPI) ecosystem. This dataset captures install and post-install-time traces from 14,271 Python packages, of which 7,127 are malicious. ML analysis using the QUT-DV25 dataset identified four malicious PyPI packages previously labeled as benign.

**Dataset Code Accessibility:**

Partly

**Dataset Code Comments:**

The dataset production procedure could be more detailed. Some pieces of code given do not match the paper content.

**Ethical Considerations:**

No, there are no or only very minor ethics concerns

**Final Justification:**

The quality of the submission has been improved. The authors have added test with DL models.

**Limitations Weaknesses:**

1. Reproducibility:
The dataset production procedure could be more detailed. Some pieces of code given do not match the paper content.

2. More tests on a broader class of methods could be done to better support the usefulness of the proposed dataset. For example,
currently, only traditional ML methods such as RF, SVM, DT and GB are tested. It would be interesting to see the performances
of deep learning based models such as MLP, CNN or RNN models.

**Strengths Contributions:**

The dataset designed to support and advance research on detecting and mitigating supply chain attacks within the Python Package Index (PyPI) ecosystem has supported ML to effectively detect four malicious PyPI packages. It outperforms reactive, metadata, and static datasets.

---

> ### Author Rebuttal · Authors · 2025-07-30
>
> #### We sincerely thank the reviewer for your valuable time, constructive feedback, and thoughtful evaluation. We sincerely appreciate the recognition of the novelty, practical value, and technical rigor of our work. Your insights are invaluable, and we address the concerns and suggestions in detail below.
>
> ---
>
> #### **Q1: The dataset production procedure could be more detailed. Some pieces of code given do not match the paper content.**
>
> #### A1: Thank you for this valuable comment. We acknowledge the importance of providing a detailed and consistent description of the dataset production procedure. To address this, we will thoroughly review and update the code repository to ensure full alignment with the methodology described in Section 3.2 of the paper. Specifically, we plan to restructure the GitHub repository as follows:
>
> ```
> QUT-DV25/
> ├── Phase (i) Dataset Collection/
> │   ├── QUT-DV25_Step 1 Malicious Package Info/
> │   ├── QUT-DV25_Step 2 Malicious Packages Info From Different Sources/
> │   ├── QUT-DV25_Step 3 Similarity Algorithms Implementation/
> │   ├── QUT-DV25_Step 4 Counterpart Benign Packages/
> ├── Phase (ii) Dataset Labeling and Validation/
> │   ├── QUT-DV25_Step 1 Malicious Validation Report and Labeling/
> │   ├── QUT-DV25_Step 2 Benign Validation Report and Labeling/
> │   ├── QUT-DV25_Step 3 Malicious Packages Metadata and Static Dataset/
> ├── Phase (iii) Trace Extraction/
> │   ├── QUT-DV25_Step 1 Isolated Env with RPIs/
> │   │   ├── *Environment Setup with eBPF and Linux Guidelines.pdf             //will be added
> │   ├── QUT-DV25_Step 2 Malicious Packages Traces/
> │   ├── QUT-DV25_Step 3 Benign Packages Traces/
> ├── Technical Validation and Benchmarks/
> │   ├── QUT-DV25_Step 1 Traces Preprocessing-Feature Engineering/
> │   ├── QUT-DV25_Step 2 Traditional ML Implementation/
> │   ├── QUT-DV25_Step 3 DL Implementation/
> │   ├── *Run Procedure for Model Training and Evaluation.pdf                  //will be added
>
> ```
> #### **Note:** Files marked with an asterisk (*) will be added to the repository.
>
> #### To further enhance reproducibility and clarity, we will add a comprehensive environment setup procedure in the dataset’s GitHub repository under “Phase (iii) Trace Extraction”, including a new file titled “Environment Setup with eBPF Program and Linux Kernel Installation Guidelines.pdf”.
>
> #### In addition, a document titled “Run Procedure for Model Training and Evaluation.pdf” will be added under the “Technical Validation and Benchmarks” directory. These two documents will detail the OS version, Raspberry Pi image specifications, environment configuration requirements, and step-by-step instructions for executing the pipeline.
>
> #### Furthermore, we will also update the repository’s README.md file with two new sections:
> - #### Environment Setup Instructions
> - #### Run Instructions
>
> #### These README sections will clearly outline system prerequisites, software dependencies, folder structure, and command-line instructions required to execute the full pipeline from end to end.
>
> #### All updates will be made publicly available in the GitHub repository. We will also revise the manuscript accordingly to reference these additions, ensuring consistency between the paper and the released code repository.
>
> ---
>
> #### **Q2: More tests on a broader class of methods could be done to better support the usefulness of the proposed dataset. For example, currently, only traditional ML methods such as RF, SVM, DT and GB are tested. It would be interesting to see the performances of deep learning based models such as MLP, CNN or RNN models.**
>
> #### A2: Thanks for this insightful suggestion. In addition to traditional ML models, we also experimented with several DL architectures, including CNN and a baseline Transformer model. These models were trained and evaluated for up to 200 epochs using the CombinedTraces input. The CNN model achieved its best performance with early stopping at epoch 72, reaching a test accuracy of 95.25% and a loss of 0.2460. Across epochs, the confusion matrix and other evaluation metrics remained largely stable, suggesting limited performance gains from extended training or the use of more complex architectures. We will incorporate these findings into the camera-ready version of the paper.
>
> #### In addition, we will include the corresponding results, logs, and performance summaries in the dataset’s GitHub repository under the following planned structure:
>
> ```
> Technical Validation and Benchmarks/
> ├── QUT-DV25_Step 1 Traces Preprocessing-Feature Engineering/
> ├── QUT-DV25_Step 2 Traditional ML Implementation/
> ├── *QUT-DV25_Step 3 DL Implementation/                                    //will be added
> └── Run Procedure for Model Training and Evaluation.pdf
> ```
>
> #### **Note:** **QUT-DV25_Step 3 DL Implementation/* is the directory that will contain the DL model details.
>
> #### Furthermore, we conducted a comparative evaluation based on model complexity, training and testing efficiency, and practical deployability in real-world security systems. The table below summarizes key metrics across the selected traditional ML and DL models:
>
> | Model           | Test Accuracy | Training Time (s) | Validation Time (s) | Test Time (s) |
> | --------------- | ------------- | ----------------- | ------------------- | ------------- |
> | RF              | 0.959941      | 0.494045          | 0.151228            | 0.118388      |
> | DT              | 0.940215      | 6.305851          | 0.041311            | 0.038888      |
> | SVM             | 0.952826      | 35.150900         | 0.437460            | 0.415141      |
> | GB              | 0.941149      | 46.963030         | 0.027708            | 0.025581      |
> | **CNN**         | **0.952518**  | **5155.121**      | **0.695444**        | **0.572358**  |
> | **Transformer** | **0.945077**  | **5532.741**      | **0.774360**        | **0.650576**  |
>
> #### While CNN and Transformer models achieved competitive accuracy, they required significantly higher training and testing time, which poses challenges for real-world deployment in security-critical environments. Although the performance metrics were comparable, the traditional ML models offered better efficiency and ease of deployment. Based on this trade-off, we selected these four traditional ML models as the primary benchmarking models.

---

> > ### Comment · Reviewer_hVc6 · 2025-08-07
> >
> > Thanks the authors for their repsonse and efforts to improve the quality of the submission. I would like to increase my score.

---

> > > ### Author Response · Authors · 2025-08-08
> > >
> > > #### We sincerely thank the reviewer for accepting our response.

---

### Official Review · Reviewer_4Qxj · 2025-07-19

**Rating:** 5
**Confidence:** 4

**Summary:**

The submission introduces QUT-DV25, a large-scale dynamic analysis dataset specifically designed to support research on detecting and mitigating advanced software supply chain attacks in the Python Package Index (PyPI) ecosystem. Recognizing the limitations of existing static and metadata-based datasets—such as their inability to detect multi-stage threats like dynamic payload generation and remote access activation—the authors construct QUT-DV25 by dynamically analyzing 14,271 PyPI packages (7,127 malicious) in isolated environments using eBPF-based kernel and user-space tracing. Each package is monitored during install and post-install phases, capturing 36 features across categories including system calls, TCP activity, file access, and behavioral patterns. The result is a unique and practical dataset that captures real-time execution behaviors not visible through conventional analysis techniques.

The study not only details the construction and validation of the dataset but also provides baseline evaluations using four machine learning models (Random Forest, Decision Tree, SVM, Gradient Boosting). These evaluations demonstrate that QUT-DV25 significantly outperforms existing datasets in detecting covert or delayed malicious behaviors, achieving up to 96% accuracy and F1-scores when using combined dynamic features. Notably, the models trained on QUT-DV25 successfully identified four malicious packages previously labeled benign, reinforcing its real-world impact. The dataset, code, and documentation are openly available, with ethical considerations addressed through isolated testing and institutional review. QUT-DV25 stands as a valuable and timely contribution to the cybersecurity and machine learning communities, offering a robust benchmark for dynamic malware detection in software supply chains.

**Additional Feedback:**

Currently, the post-install execution trace is limited to 120 seconds, which may miss delayed or time-triggered malicious behaviors.

Suggestion: Future iterations could include either configurable or longer trace windows, or event-driven trace extensions (e.g., based on process activity or user simulation triggers).

The dataset is specific to the PyPI ecosystem, which is understandable for a first release.

Suggestion: Consider generalizing the data collection and trace framework to support NPM, RubyGems, or Maven, enabling broader applicability and comparisons across ecosystems.


Many packages behave differently once imported or invoked (e.g., CLI tools or frameworks).

Suggestion: Extend the behavioral simulation to include import-time or runtime function execution, potentially using fuzzing tools or scripted inputs to simulate user interaction.

While the ML models perform well, it would be insightful to understand why—which trace categories contribute most to classification?

Suggestion: Incorporate interpretability techniques (e.g., SHAP, LIME, permutation importance) to identify which syscall patterns or file access types most strongly signal malicious behavior.


How does the dataset handle benign packages with occasional anomalous behaviors? For instance, tools that use shell access or open ports for legitimate purposes.

Is there a mechanism for community contribution or labeling correction? Given the dynamic nature of malware discovery, a versioned or community-augmented dataset may be valuable.

Can the testbed be modularized for reproducibility by others? Is there Docker or script-based provisioning for the isolated environments used in trace collection?

**Dataset Code Accessibility:**

Yes

**Dataset Code Comments:**

The submission provides open, well-documented, and fully executable access to both the dataset and supporting code, meeting NeurIPS standards for reproducibility and accessibility.

The full dataset is publicly hosted at Harvard Dataverse DOI: 10.7910/DVN/LBMXJY.

A complete list of analyzed Python packages is also available via a dedicated web portal: https://qut-dv25.dysec.io.

The dataset includes 14,271 PyPI packages (7,127 malicious), with extracted dynamic behavioral traces and labels.

**Ethical Comments:**

No additional ethical review is needed.

- All packages analyzed were publicly available from the Python Package Index (PyPI) and external threat intelligence sources (e.g., GitHub advisories, malware databases)

- No personally identifiable information (PII), user-generated content, or sensitive private data is included in the dataset, minimizing privacy risks.

- All dynamic analysis was performed in isolated, controlled sandbox environments, with network containment and monitoring via eBPF. This ensures that malicious behaviors were fully contained and did not affect external systems or networks.

- The authors acknowledge dual-use risks (e.g., misuse of eBPF-based monitoring), but contextualize them within controlled research environments and offer mitigation strategies.

**Ethical Considerations:**

No, there are no or only very minor ethics concerns

**Limitations Weaknesses:**

Deployed using 16 Raspberry Pi devices with kernel-level eBPF support to simulate realistic execution environments (Section 3.1, Figure 1).

The dataset was collected exclusively in isolated environments using 16 Raspberry Pi devices running Ubuntu 24.4 LTS and kernel v6.8.0-1012-raspi with eBPF instrumentation.

This setup, while controlled, may not reflect real-world diversity in hardware architectures, OS versions, Python environments, or dependency configurations.

Many attacks are environment-sensitive (e.g., payloads that activate only under specific OS conditions or hardware features).

The tracing duration is fixed at 120 seconds for post-install-time behavior. :-Sophisticated malware often uses time delays or staged activation that may occur after the initial 2-minute window. This cutoff could miss covert or dormant malicious behaviors, limiting the dataset’s utility for detecting long-term or time-triggered threats.

Demonstrates that combined dynamic features outperform static and metadata datasets in accuracy (95.99%) and F1-score (96.02%), highlighting the dataset's relevance and value.

Imbalanced or Noisy Behavioral Signals - The dataset targets only the Python Package Index (PyPI).:-Software supply chains encompass other major ecosystems (e.g., NPM, Maven, RubyGems). PyPI-specific features (e.g., setup.py, whl files) are not generalizable to other ecosystems with different install workflows or file structures.

Although the eBPF traces are collected in clean environments, noise in dynamic execution is still a concern. :- Traces may include benign signals unrelated to the malicious payload, which could confuse learning algorithms. High-dimensional feature vectors (36 trace types) may contain correlated or redundant features, affecting model interpretability and efficiency.

Absence of Delayed or User-Triggered Behavior Simulation - The dataset focuses on install and post-install time, but does not simulate user interaction or runtime behaviors.:-Certain malware activates only in response to user input, command-line options, or runtime context (e.g., loading a specific module). This creates blind spots in detecting interaction-triggered malicious activity.

**Strengths Contributions:**

Scope: 14,271 Python packages, including 7,127 malicious samples, captured via dynamic install-time and post-install-time execution in eBPF-instrumented sandbox environments.

Distinction: Existing datasets are static or metadata-based and miss install-time behaviors. QUT-DV25 uniquely captures 36 dynamic behavioral features across six categories (e.g., system calls, TCP flows, file access patterns)

Novel Capability: Identified 4 previously undetected malicious packages and contributed to their removal from PyPI.

Clear Comparison with prior datasets (Metadata [20], Static [22], Hybrid [23]), showing that those datasets fail to detect remote access, dynamic payloads, and post-install behavior.

---

> ### Author Rebuttal · Authors · 2025-07-30
>
> #### We sincerely thank the reviewer for your valuable time, constructive feedback, and thoughtful evaluation. We sincerely appreciate the recognition of the novelty, practical value, and technical rigor of our work. Your insights are invaluable, and we address the concerns and suggestions in detail below.
>
> ---
>
> #### **Q1: Deployed using 16 Raspberry Pi devices with kernel-level eBPF support to simulate realistic execution environments (Section 3.1, Figure 1). The dataset was collected exclusively in isolated environments using 16 Raspberry Pi devices running Ubuntu 24.4 LTS and kernel v6.8.0-1012-raspi with eBPF instrumentation. This setup, while controlled, may not reflect real-world diversity in hardware architectures, OS versions, Python environments, or dependency configurations. Many attacks are environment-sensitive (e.g., payloads that activate only under specific OS conditions or hardware features).**
>
> #### A1: Thanks for highlighting the limitations of our homogeneous testbed environment. We completely agree that our current setup may not capture environment-sensitive attacks due to the lack of hardware, OS, and dependency diversity. However, this design prioritized safety, reproducibility, and cost-efficiency for controlled malicious package execution. We employed eBPF for trace collection during installation and post-installation phases, which is a powerful and secure kernel-level instrumentation tool. While full support for eBPF is currently available only on Linux, our approach remains well representative of any Linux-based system, including widely used server and desktop distributions.
>
> #### Although eBPF for Windows is a promising development, its initial release does not support comprehensive system call tracing. Concurrently, we also explored alternative tools for Windows, such as Wireshark and Sysdig. However, these tools lack customizable programmable flexibility and secure isolation unless kernel-level modifications are applied, which introduces significant safety and feasibility concerns.
>
> #### However, our methodology is not limited to a specific hardware or operating system. It uses both kernel and user-level probes to monitor system behavior during package installation and post-installation phases, making it conceptually applicable across different OS environments. While expanding to Windows and macOS introduces logistical and ethical challenges, we consider this an important direction for future work and will acknowledge this limitation in the camera-ready version of the paper.
>
> ---
>
> #### **Q2: The tracing duration is fixed at 120 seconds for post-install-time behavior. Sophisticated malware often uses time delays or staged activation that may occur after the initial 2-minute window. This cutoff could miss covert or dormant malicious behaviors, limiting the dataset’s utility for detecting long-term or time-triggered threats.**
>
> #### A2: Thanks for this insightful observation. In theory, dormant malicious code can delay execution indefinitely, making it infeasible to observe open-ended run-time behavior in a practical experimental setting. This presents a fundamental challenge: any fixed time threshold may be bypassed by attackers who intentionally delay malicious actions.
>
> #### To address this, our focus was on identifying malicious behaviors that manifest shortly after installation-consistent with many real-world supply chain attacks that aim to execute during or soon after package setup. To empirically evaluate an appropriate time window, we analyzed 64 high-risk malicious packages. These packages had previously exhibited suspicious behaviors such as system freezing, infinite waiting, and version looping, which were strong indicators of potential delayed execution. For these packages, we collected dynamic traces using four different observation windows: 60, 120, 300, and 600 seconds, within a controlled and isolated environment.
>
> #### Critically, no significant divergence in directory access, network activity, or system call patterns was observed beyond the 120-second mark. Based on this analysis, we adopted the 120-second threshold as a practical compromise, providing sufficient behavioral coverage beyond our 60-second baseline. However, we acknowledge that some highly evasive threats may employ longer execution delays.
>
> #### We will highlight this limitation in *Section 5* of the camera-ready version and consider extending the observation window as an important direction for future work.
>
> ---
>
> #### **Q3: Demonstrates that combined dynamic features outperform static and metadata datasets in accuracy (95.99%) and F1-score (96.02%), highlighting the dataset's relevance and value.**
>
> #### A3: We sincerely appreciate the reviewer’s important observation. Our results demonstrate that the combined traces feature set significantly outperforms static and metadata-based approaches. This improvement is due to the dynamic features’ ability to capture real-world attack patterns. These include multiphase malware execution, remote access activation, dynamic payload generation, and immediate post-installation behaviors, which often evade detection by static analysis or metadata inspection. This performance highlights the unique value of our dataset. It enables ML models to identify novel, evasion-capable threats by capturing behavioral fingerprints during both the installation and post-installation phases.
>
> ---
>
> #### **Q4: The dataset targets only the Python Package Index (PyPI). Software supply chains encompass other major ecosystems (e.g., NPM, Maven, RubyGems). PyPI-specific features (e.g., setup.py, whl files) are not generalizable to other ecosystems with different install workflows or file structures.**
>
> #### A4: Thank you for this insightful observation. We acknowledge that the current scope of the dataset is focused on the PyPI ecosystem. We selected PyPI because Python is widely used in data science, machine learning, and automation. Consequently, these domains have increasingly attracted the attention of malicious actors in recent years.
>
> #### However, our methodology was designed as a modular and extensible framework for analyzing behavioral traces during both the installation and post-installation phases of software packages. The core components of the framework include malicious package collection, benign counterpart selection, dataset labeling, validation, and trace extraction. These components are adaptable to other software ecosystems with minimal adjustments.
>
> #### Importantly, our approach captures system-level behaviors using both kernel and user-level probes, making it independent of ecosystem-specific packaging formats or structures. While ecosystems such as PyPI, NPM, and Maven differ in setup files and installation workflows, adapting our approach only requires modifying the execution instructions, not reengineering the eBPF setup or the isolated environment.
>
> #### We agree that expanding to additional ecosystems will significantly enhance the impact and utility of our dataset, and we consider this an important direction for future research, as mentioned in *Section 5* of the paper.
>
> ---
>
> #### **Q5: Although the eBPF traces are collected in clean environments, noise in dynamic execution is still a concern. Traces may include benign signals unrelated to the malicious payload, which could confuse learning algorithms. High-dimensional feature vectors (36 trace types) may contain correlated or redundant features, affecting model interpretability and efficiency.**
>
> #### A5: Thank you for highlighting this important concern. While the eBPF traces were collected in clean environments, we acknowledge that benign signals unrelated to malicious payloads may persist and potentially affect model learning. To address this, we conducted a correlation analysis to ensure feature independence and retained only those traces that represented distinct behavioral patterns. Specifically, from an initial set of 62 features, we removed 26 due to high correlation and informational redundancy. This resulted in 36 relevant and unique features, enhancing both model interpretability and efficiency.
>
> ---
>
> #### **Q6: The dataset focuses on install and post-install time, but does not simulate user interaction or runtime behaviors. Certain malware activates only in response to user input, command-line options, or runtime context (e.g., loading a specific module). This creates blind spots in detecting interaction-triggered malicious activity.**
>
> #### A6: We thank the reviewer for this insightful observation. We agree that our current tracing scope, which focuses on installation and immediate post-installation activities, does not capture the full spectrum of malware behaviors. In particular, it may overlook actions triggered by user interaction, command-line arguments, or specific runtime conditions. We acknowledge this as an important limitation and will highlight it in *Section 5* of the camera-ready version. Exploring techniques to simulate or monitor such runtime triggers represents a valuable and promising direction for future work.
>
> #### Finally, we sincerely thank the reviewer for their thoughtful and constructive feedback, which offers valuable suggestions and important directions to enhance the dataset’s robustness, generalizability, and usability.

---

> > ### Author Response · Authors · 2025-08-08
> >
> > #### Dear Reviewer,
> >
> > #### Thank you for your valuable time, constructive feedback, and thoughtful evaluation of our submission. In our rebuttal, we have addressed all of your valuable comments. Please let us know if any further clarification is required.
> >
> > #### Looking forward to your feedback.

---

### Decision · Program_Chairs · 2025-09-18

**Decision:**

Accept (poster)

**Comment:**

In this submission, the authors focused on the supply chain attack of software and developed a dynamic analysis dataset designed to detect advanced PyPI malware. Considering the fact that the current software attack detection datasets mainly focus on static attacks and ignore the supply chain attacks in the installation phase, the proposed dataset fills the blank for the community. Although one reviewer has a concern about the attractiveness of this dataset for the ML/AI community, I don't think this is a big issue --- actually, I think the dataset is interesting and helps build connections between ML and software engineering. Therefore, I tend to accept this work.